# Fate mapping analysis reveals a novel murine dermal migratory Langerhans-like cell population

Jianpeng Sheng[1,2], Qi Chen[2], Xiaoting Wu[2], Yu Wen Dong[2], Johannes Mayer[3], Junlei Zhang[1], Lin Wang[1], Xueli Bai[1], Tingbo Liang[1], Yang Ho Sung[2], Wilson Wen Bin Goh[2], Franca Ronchese[3], Christiane Ruedl[2]*

[1]Zhejiang Provincial Key Laboratory of Pancreatic Disease, The First Affiliated Hospital, Zhejiang University School of Medicine, Hangzhou, China; [2]Nanyang Technological University, School of Biological Sciences, Singapore, Singapore; [3]Malaghan Institute of Medical Research, Wellington, New Zealand

**Abstract** Dendritic cells residing in the skin represent a large family of antigen-presenting cells, ranging from long-lived Langerhans cells (LC) in the epidermis to various distinct classical dendritic cell subsets in the dermis. Through genetic fate mapping analysis and single-cell RNA-sequencing, we have identified a novel separate population of LC-independent CD207$^+$CD326$^+$ LC$^{like}$ cells in the dermis that homed at a slow rate to the lymph nodes (LNs). These LC$^{like}$ cells are long-lived and radio-resistant but, unlike LCs, they are gradually replenished by bone marrow-derived precursors under steady state. LC$^{like}$ cells together with cDC1s are the main migratory CD207$^+$CD326$^+$ cell fractions present in the LN and not, as currently assumed, LCs, which are barely detectable, if at all. Cutaneous tolerance to haptens depends on LC$^{like}$ cells, whereas LCs suppress effector CD8$^+$ T-cell functions and inflammation locally in the skin during contact hypersensitivity. These findings bring new insights into the dynamism of cutaneous dendritic cells and their function opening novel avenues in the development of treatments to cure inflammatory skin disorders.

*For correspondence:
ruedl@ntu.edu.sg

Competing interests: The authors declare that no competing interests exist.

## Introduction

In 1868, Paul Langerhans described a novel dendritic-shaped, non-pigmentary cell population in the epidermis (*Langerhans, 1868*). These so-called Langerhans cells (LCs) were first classified as cellular members of the nervous system, due to their morphological similarity with neurons. It was not until the 1980s when it became clear that this peculiar epidermal cell fraction with its potent antigen presentation activity belonged to the dendritic cell (DC) family (*Romani and Schuler, 1989*; *Schuler and Steinman, 1985*). Despite the fact that LCs share many features with DCs, they are generally considered as epidermal tissue-resident macrophages, mainly due to their dependence on CSF1, their embryonic origin and local self-maintenance (*Wynn et al., 2013*), although a conventional 'macrophage signature' (e.g. CD16/32, CD64, and MerTK expression) is missing (*Gautier et al., 2012*).

LCs can sense invading pathogens and initiate an intrinsic maturation process that drives their migration out of the epidermis (*Romani et al., 2001*). As such, LCs have been regarded as a prototype antigen-presenting cell (APC) (*Nagao et al., 2009*) that can, after antigen capture, migrate to the draining lymph nodes (LNs) to initiate an immune response by priming naïve LN-resident T-cells (*Romani et al., 2003*). Antigen presentation can, however, occur in skin-draining LNs independently of LCs (*Henri et al., 2010*). In fact, the skin hosts several other distinct dermal DC subpopulations (*Henri et al., 2010*; *Kissenpfennig et al., 2005*), the presence of which complicates the analysis of the cellular contribution to skin immune responses, such as contact hypersensitivity (CHS).

**eLife digest** Our immune cells are constantly on guard to defend and protect us against invading pathogens, such as bacteria and viruses. Specialized immune cells, known as antigen-presenting cells, or APCs, have a key role in this process. They engulf invaders, chew them up, and travel to the closest local lymph node to stimulate other immune cells with small fragments of these pathogens. This ramps up the immune response to control infection and disease.

APCs are a large and diverse family of immune cells, which includes dendritic cells and macrophages. Some APCs work as mobile surveillance units, travelling around the body to find new threats. Others embed themselves in particular organs and tissues, such as the skin, to provide local, on-the-spot surveillance. Langerhans cells are one of the main types of APC in the skin and are found in the thin outer layer of the epidermis. While it is commonly believed that Langerhans cells can move from the epidermis to the skin-draining lymph nodes, some seemingly contradictory evidence exists to suggest that this may not be the case.

Now, Sheng et al. have investigated this issue by tracking APCs, including Langerhans cells, in the skin of mice. A powerful genetic cell labelling technique allowed them to track the movement of immune cells inside a living mouse. Sheng et al. found that majority of 'real' Langerhans cells did not leave the skin. Yet, a second lookalike cell that shared many of the same features of a Langerhans cell was found in the dermal layer of skin, and this cell could travel to local lymph nodes. Both the original and lookalike cells had distinct and separate roles in the skin.

This research, which has uncovered a new type of Langerhans-like immune cell in the skin, may be extremely useful for developing new targeted therapies to boost immune responses during infection; or to suppress inappropriate immune activation that can lead to autoimmune diseases, such as psoriasis.

Consequently, the paradigm of 'who is doing what' (i.e. epidermal LCs versus dermal DC counterparts) is still controversial (*Bennett et al., 2005*; *Bobr et al., 2010*; *Bursch et al., 2007*; *Clausen and Stoitzner, 2015*; *Kaplan et al., 2005*; *Noordegraaf et al., 2010*; *West and Bennett, 2017*).

Here, we demonstrate that under steady-state conditions, LCs most likely do not exit the skin, or if so, in very low numbers. Through a combined use of genetic fate mapping and novel inducible LC-ablating mouse models, we show that the originally described LN LC fraction is actually an independent LC$^{like}$ cell population that originates from the dermis, not from the epidermis. These LC$^{like}$ cells are ontogenitically different from LCs and are replaced over time by bone marrow (BM)-derived cells with slow kinetics before trafficking to the LN.

## Results

### LC$^{like}$ cells are found in dermis and LNs

The skin and the skin-draining LNs contain several distinct DC subpopulations. To delineate migratory LCs and dermal DCs, we profiled DC subsets in the epidermis, dermis, and skin-draining LNs. In the epidermis, we confirmed that CD326$^+$CD207$^+$ LCs are predominantly found within the CD11b$^{hi}$F4/80$^{hi}$ fraction (*Nagao et al., 2009*; *Valladeau et al., 2000*; *Figure 1A*). In the dermis, we found a fraction of CD11b$^{hi}$F4/80$^{hi}$ cells that co-expressed CD326 and CD207 (*Figure 1B*, upper panel). These cells could be immigrated LCs, although we cannot exclude a contamination from the epidermis during the isolation procedure. As expected, the remaining dermal CD11b$^{hi}$F4/80$^{hi}$ cells were CD326$^-$CD207$^-$ tissue-resident macrophages (*Sheng et al., 2015*; *Tamoutounour et al., 2013*). Dermal DCs were localized in the F4/80$^{int}$ and CD11c$^{hi}$MHCII$^+$ DC fraction, which we could separate into three subpopulations based on CD103 and CD11b expression: CD103$^+$CD11b$^-$ (defined as cDC1), CD103$^-$CD11b$^{low}$, and CD103$^-$CD11b$^{hi}$ (defined as CD11b$^{hi}$). CD103$^+$CD11b$^-$ DCs but not CD103$^-$CD11b$^{hi}$ DCs co-expressed CD326 and CD207. We could also divide the CD103$^-$CD11b$^{low}$ subpopulation into CD326$^-$CD207$^-$ (defined as triple negative [TN]) and CD326$^+$CD207$^+$ (defined as LC$^{like}$) fractions (*Figure 1B*, right panel). The signal regulatory protein α (Sirpa), a myeloid cell-specific receptor, was expressed on dermal LCs, LC$^{like}$, TN, and CD11b$^{hi}$ DCs, while, as expected,

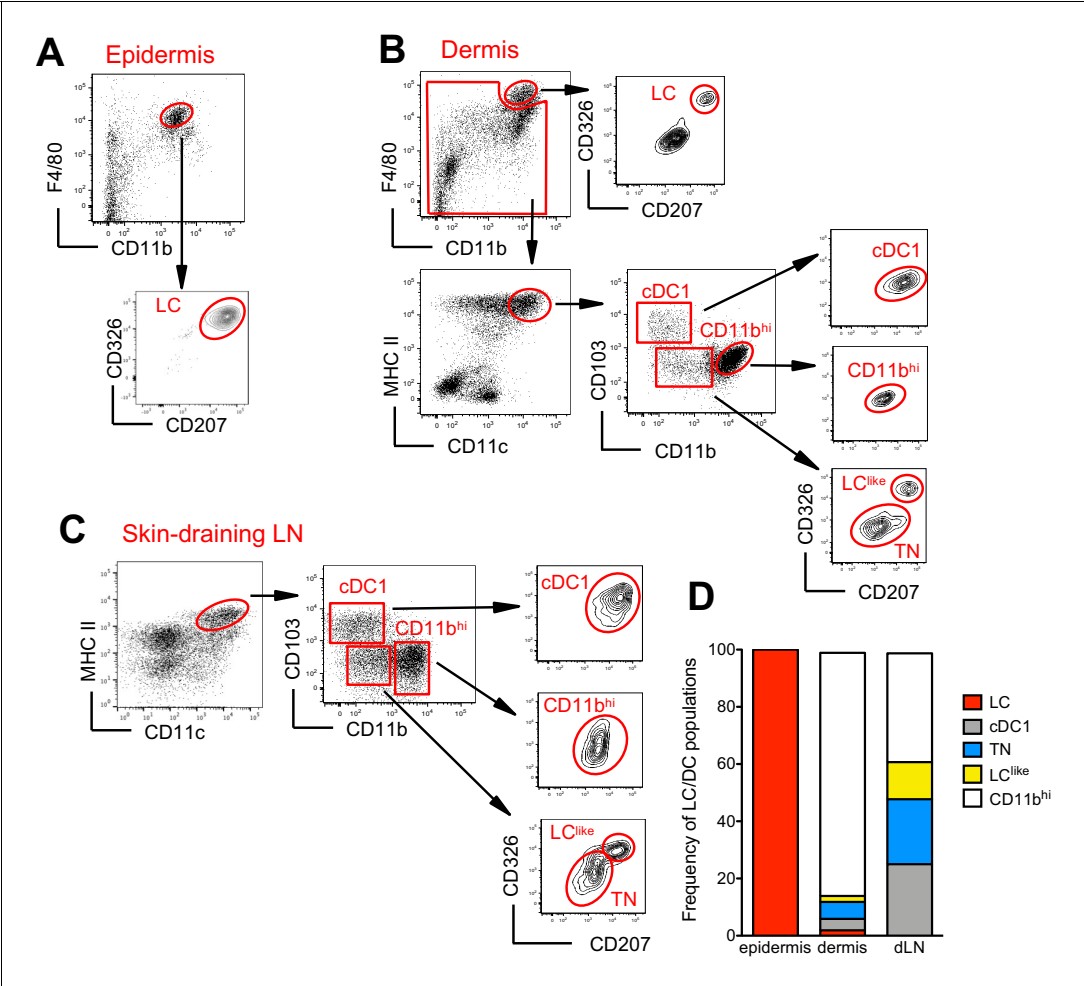

**Figure 1.** Characterization of cutaneous Langerhans cell (LC) and dendritic cell (DC) subpopulations. (**A**) Representative flow cytometry dot plots for LC characterization in the epidermis. Cells from epidermis were first gated for FSC, SSC, and CD45 (not shown). Then, CD45+ cells were analysed for CD11b and F4/80 expression. The CD11bhiF4/80hi cell fraction was further analysed for CD207 and CD326 expression to identify classical bona fide LCs. (**B**) Representative flow cytometry dot plots for dermal LC and DC subpopulations. Isolated dermis cells were first gated for FSC, SSC, and CD45 (not shown). CD45+ cells were then analysed for CD11b and F4/80 expression. The CD11bhi F4/80hi fraction contained classical CD207+CD326+ LCs. The remaining cells were gated for CD11c+MHC II+ DCs and separated into three subsets based on CD103 and CD11b expression: CD103+CD11b- cells (labelled cDC1), CD103-CD11bhi DCs (labelled CD11bhi), and CD11blow/neg. CD207 and CD326 expression was detectable on cDC1 but not CD11bhi DCs, whereas CD11blow cells were further separated into CD207-CD326- (labelled triple negative [TN]) and CD207+CD326+ (labelled LClike). (**C**) Representative flow cytometry dot plots for cutaneous DC subpopulations in auricular skin-draining LNs. LN cells were first gated for FSC and SSC to exclude small lymphocytes before F4/80 and CD11b staining. The cell fraction excluding F4/80hi/CD11bhi cells was separated by CD11c and MHCII. CD11chiMHCIIhi migratory DCs were gated and analysed for CD103, CD11b, CD207, and CD326 expression. Four subsets were detected: CD103+CD11b-CD207+CD326+ (cDC1), CD103-CD11blowCD207-CD326- (TN), CD103-CD11blowCD207+CD326+ (LClike), and CD103-CD11bhiCD207-CD326-/+(CD11bhi). (**D**) Frequency of each DC subpopulation (LC, cDC1, LClike, TN, and CD11bhi) present in epidermis, dermis, and cutaneous lymph node (LN), respectively.

The online version of this article includes the following figure supplement(s) for figure 1:

**Figure supplement 1.** Expression profiles of myeloid marker signal regulatory protein α (Sirpa) and costimulatory molecules (CD80 and CD86) (**A**) and DC-SIGN (CD209a) (**B**).

**Figure supplement 2.** Flow cytometry analysis and characterization of EGFP+ cells in the epidermis (upper panel), dermis (middle panel), and skin-draining lymph nodes (LNs) (lower panel) obtained from the Lang-EGFP mouse.

dermal CD103+ DCs lacked this receptor (*Figure 1—figure supplement 1A*), result validated also by the single-cell RNA-sequencing (scRNA-seq) analysis shown later (*Figure 3—figure supplement 1*). In terms of costimulatory receptors, dermal LCs and LClike cells express similar levels of CD80

and CD86, whereas the remaining DC subpopulations display lower levels (*Figure 1—figure supplement 1A*).

To track the corresponding migratory DCs in the skin-draining LNs, we first gated on CD11c$^{int-hi}$MHCII$^{hi}$ cells, which represent the migratory DC fraction (*Sheng et al., 2017*). Similar to our findings in the dermis, CD11b and CD103 labelling separated the migratory DCs into CD103$^+$CD11b$^-$ (cDC1), CD103$^-$CD11b$^{hi}$ (CD11b$^{hi}$), and CD103$^-$CD11b$^{low}$ cells (*Figure 1C*). The CD103$^-$CD11b$^{low}$ cells could be further separated into two fractions: CD326$^-$CD207$^-$ (TN) and CD326$^+$CD207$^+$ (LC$^{like}$) subpopulations (*Figure 1C*). Notably, we did not detect the bona fide epidermal and dermal LCs showing the original F4/80$^{hi}$CD11b$^{hi}$ phenotype in the LN (*Figure 1C*, right, lower panel).

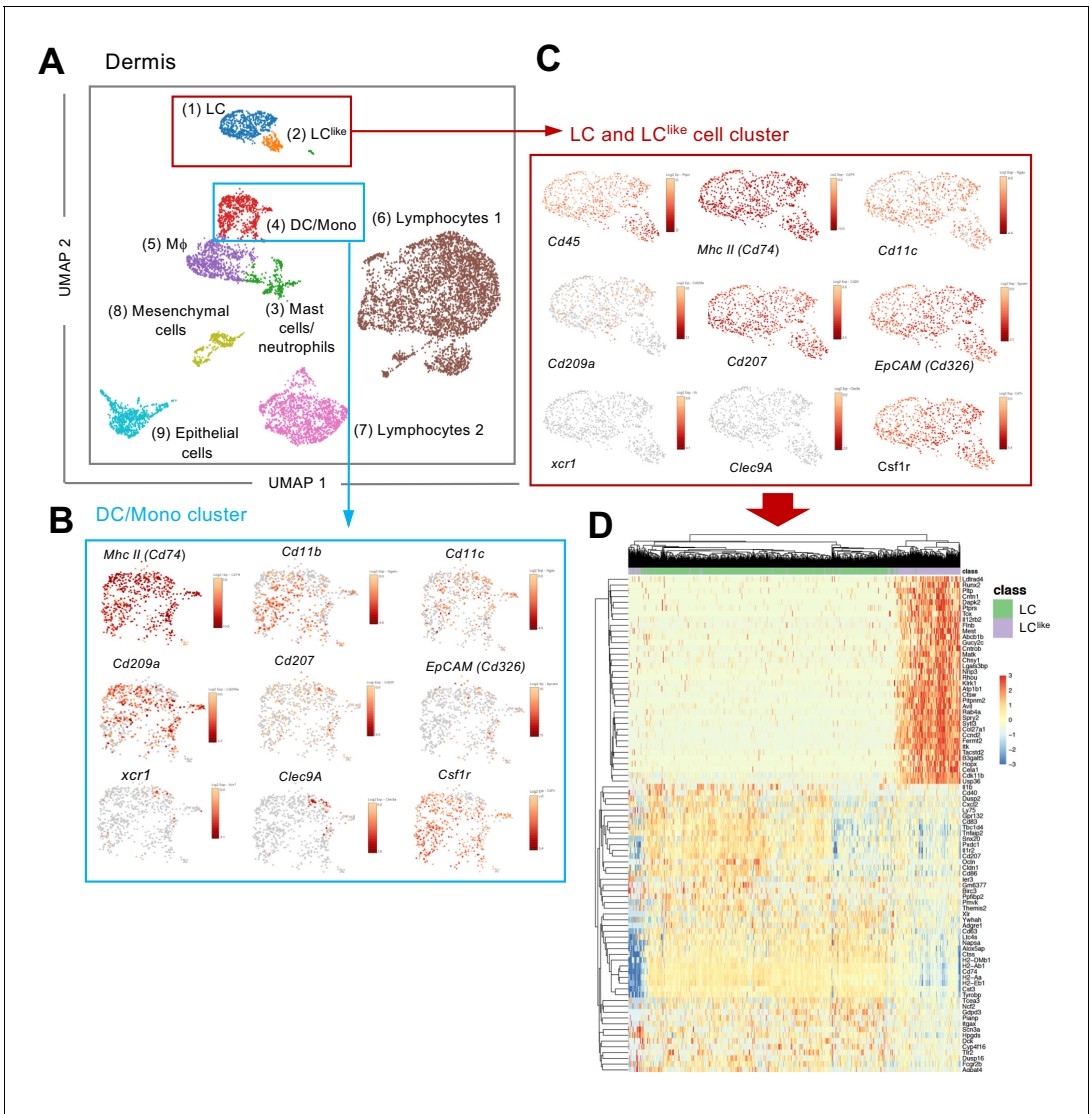

**Figure 2.** Single-cell RNA-seq analysis reveals Langerhans cell (LC) and LC$^{like}$ cells as two distinct cell populations in the dermis. 9605 cells pooled from the dermis collected from six mice which passed QC were imported for Seurat analysis. (**A**) Uniform manifold approximation and projection (UMAP) plot is revealing the existence of nine distinct cell clusters (1) LC (blue), (2) LC$^{like}$(orange), (3) mast cells/neutrophils (green), (4) dendritic cell (DC)/monocytes (red), (5) macrophages (purple), (6) lymphocytes 1 (brown), (7) lymphocytes 2 (pink), (8) mesenchymal cells (light green), and (9) epithelial cells (light blue). (**B, C**) UMAP maps showing the expression of various LC signature genes in DC/mono (**B**) and LC/LC$^{like}$ clusters (**C**). (**D**) Heat-map of single-cell gene expression data based on the top differentially expressed genes discriminating LC/LC$^{like}$ clusters. Cells (LC in green; LC$^{like}$ in purple) are shown in rows and genes in columns.

The online version of this article includes the following figure supplement(s) for figure 2:

**Figure supplement 1.** Expression of myeloid receptors and transcription factors among dermal CD45+ cells.

In agreement with previous work (*Henri et al., 2010*), the CD11b$^{hi}$ DC fraction represented the largest DC subpopulation in the dermis, whereas in the LN all four DC subpopulations (CD11b$^{hi}$, cDC1, TN, and LC$^{like}$) were almost equally represented (*Figure 1D*).

To confirm our observation, we took advantage of the Lang-EGFP mouse to trace directly EGFP-expressing CD207$^+$ cells in all three tissues (epidermis, dermis, and skin-draining LN). Clearly epidermal EGFP$^+$ cells were co-expressing high levels of CD326 and F4/80 (*Figure 1—figure supplement 2*, upper panel). In the dermis, two main EGFP$^+$ cells' populations were detectable: one with lower levels of CD326 expressing xCR1, typical marker for cDC1 and a second population co-expressing EGFP and CD326 was further separated into F4/80$^{hi}$ and F4/80$^{low}$ fractions (*Figure 1—figure supplement 2*, middle panel). Differently to the skin, EGFP expression in the LN was weaker and was restricted to CD326$^{low}$ and CD326$^{hi}$ cells: CD326$^{low}$ cells represent resident (CD11c$^{hi}$MHCII$^{int}$) DCs and CD326$^{hi}$ cells the migratory (CD11c$^{int}$MHCII$^{hi}$) fraction. With respect to resident DCs, EGFP$^+$ cells were only detectable in xCR1$^+$ cells but not in the CD11b$^+$ fraction. On the contrary, the migratory DCs were further subdivided into CD326$^{hi}$EGFP$^{+/low}$ xCR1$^+$ cDC1 and F4/80$^{low}$ LC$^{like}$ cells. No CD326$^{hi}$EGFP$^{+/low}$ F4/80$^{hi}$ cells were detectable in the LN (*Figure 1—figure supplement 2*, lower panel).

Because we detected no phenotypic F4/80$^{hi}$ LCs in the LNs, we hypothesized that the cutaneous DCs en route to the LN were not derived from epidermal LCs, but rather from distinct dermal CD11b$^{hi}$, cDC1, TN, and F4/80$^{low}$ LC$^{like}$ DC populations. This analysis cannot exclude, however, the possibility that the migrating LCs might change their phenotype as demonstrated previously (*Schuler and Steinman, 1985*).

## scRNA-seq confirms the presence of two independent LC and LC$^{like}$ cell populations in the dermis

Since LC and LC$^{like}$ cells co-exist together in the dermis, we aimed to investigate their relationship and respective gene signature by scRNA-seq analysis. Unsupervised clustering and uniform manifold approximation and projection (UMAP) were performed on 9605 enriched cells isolated from the dermis of ears obtained from five mice. The origin of distinct CD45$^+$ and CD45$^-$ dermal cell subpopulations are visualized in a colour-coded UMAP plot (*Figure 2A*). Nine different cell clusters could be broadly identified by unsupervised clustering and classified as follows: (1) LC, (2) LC$^{like}$, (3) mast cells/neutrophils, (4) DC/monocytes, (5) macrophages, (6) lymphocytes 1, (7) lymphocytes 2, (8) mesenchymal cells, and (9) epithelial cells. Conventional DCs, monocytes, and other myeloid-related signature genes, such as *Zbtb46* (DCs), *Xcr1* and *Clec9a* (cDC1), *Siglech* (plasmacytoid DC), *Ly6c* and *Ccr2* (monocytes), *Gata2* and *Fcer1a* (mast cells), and *Ly6g* (neutrophils), are mainly detectable in the DC/mono and mast cell/neutrophil clusters (3–4) and are mainly absent or weakly expressed in the LC/LC$^{like}$ clusters (1–2) (*Figure 2B,C* and *Figure 2—figure supplement 1*). *Cd207* and *Cd326* expressing cells are detected in LC (1), LC$^{like}$ (2), as well as in DC/monocyte cluster (4), which confirms the presence of three distinct CD207$^+$CD326$^+$ dermal subpopulations observed by flow cytometry (*Figure 1B*). *Cd207* and *Cd326* expressing cells detected in the cluster 4 are co-expressing *Clec9a*, *Xcr1*, *Irf8* hence they represent the cDC1s (*Figure 2B* and *Figure 2—figure supplement 1*). *Cd207* and *Cd326* expressing cells in clusters 1 (LC) and 2 (LC$^{like}$) share many of the previously reported LC signature genes (e.g. *Cd11c*, *Adgre1*, *Cd74*, *Mafb*, *Pu.1*, *Csf1r*, *Tgfbr1*) (*Figure 2C* and *Figure 2—figure supplement 1*), but several other genes are differentially expressed in LC$^{like}$ cells (e.g. *Tgfbr2*, *Sylt3*, *Col27a1*, *Fernt2*, *Spry2*) or in LC cells (e.g. *Cd209a*, *Agpat4*, *Birc3*, *Dusp16*, *Gdpd3*, *Ly75* and *Ppfibp2*), respectively (*Figure 2C,D*).

To further elucidate the relationship between different dermal DC and macrophage populations, a developmental trajectory was obtained from a UMAP analysis specifically obtained from clusters 1, 2, 4, and 5 shown in *Figure 2* (*Figure 3A*). Clearly there is a close relationship between LC and LC$^{like}$ as well as between macrophages/monocytes and CD11b$^+$/TN DCs, whereas cDC1s are identified as a separate independent cell cluster (*Figure 3A and B*). Furthermore, detailed transcription factor (TF) analysis revealed that LC and LC$^{like}$ cells share many TFs, some equally expressed (*Mafb*, *Irf4*, *Irf8*), some higher expressed in LC (*Pu.1*), and some more elevated in LC$^{like}$ (*Stat3*, *Runx2*, *Runx3*, *Id2*, *Klf4*, *Maf*) (*Figure 3C* and *Figure 3—figure supplement 1A*). *Zbtb46*, a TF selectively expressed on classical DCs, is expressed on both LC and LC$^{like}$ cells but not as high as on classical DCs (cDC1, CD11b$^+$, and TN) (*Figure 3—figure supplement 1A*). Interestingly, *Zeb2*, a specific cDC2 TF, is only weakly expressed on LC and LC$^{like}$ cells (*Figure 3—figure supplement 1A*).

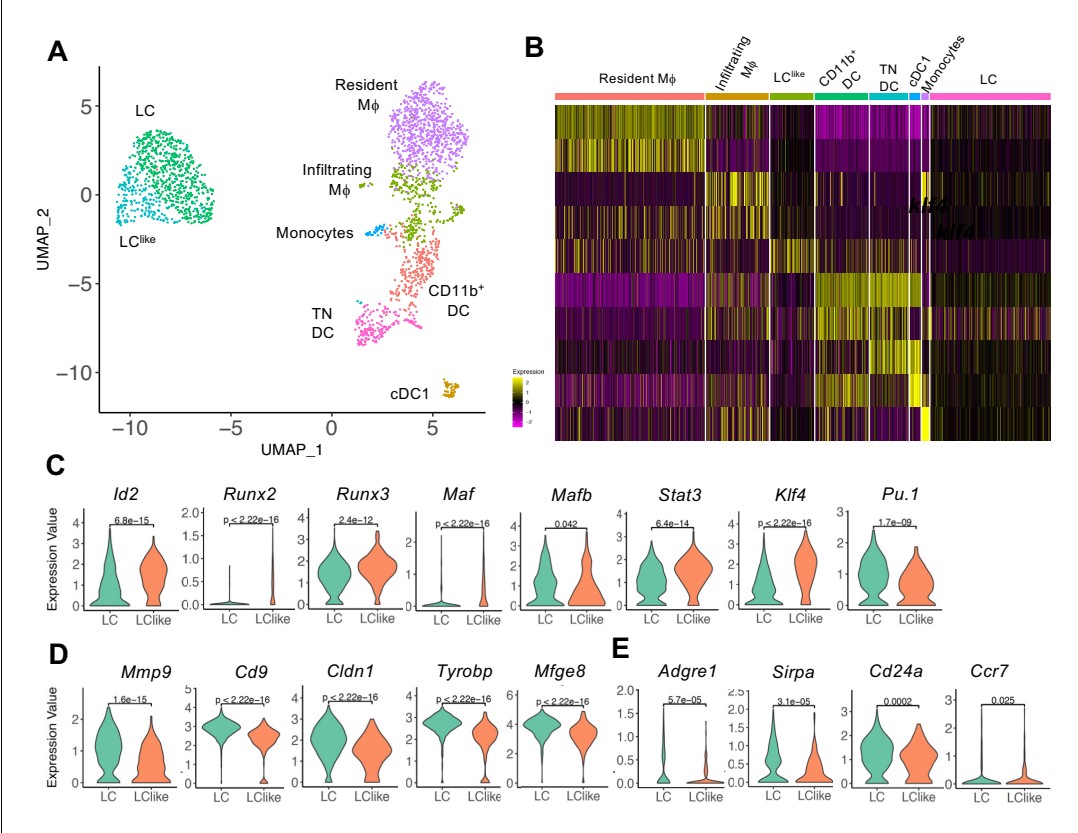

**Figure 3.** Detailed uniform manifold approximation and projection (UMAP) analysis from clusters 1, 2, 4, and 5 visualizes eight distinct Langerhans cell (LC)/dendritic cell (DC) and macrophage subpopulations. (**A**) UMAP plot showing eight distinct LC/DC and macrophage clusters: LC (emerald green), LC$^{like}$ (turquoise), cDC1 (ocher), CD11b$^+$ (red orange), triple negative (TN) DCs (magenta), resident macrophages (purple), monocytes (light blue), and infiltrating macrophages (green). Colours indicate unsupervised clustering by PhenoGraph. (**B**) Heat-map of single-cell gene expression data based on the top differentially expressed genes between the eight cell clusters. Yellow: upregulated; purple: downregulated. (**C**) Violin plots comparing transcription factor (TF) expression in LC and LC$^{like}$ cells. (**D**) Violin plots showing mRNA expression profile of LC signature genes in LC and LC$^{like}$ cells. (**E**) Violin plots showing *Adgre1* (F4/80), *Sirpa*, *Cd24a*, and chemokine receptor *Ccr7* expression in LC and LC$^{like}$ cells.

The online version of this article includes the following figure supplement(s) for figure 3:

**Figure supplement 1.** Violin plots showing mRNA expression profile of (A) transcription factor (TF) genes, (B) *Sirpa*, and (C) IL2 receptor genes (*Il2rb*, *Il2rg*) in eight distinct dermal Langerhans cell (LC), dendritic cell (DC), and macrophage subpopulations.

Furthermore, LC-related genes, such as *Mmp9, Cd9, Mfge8, Cldn1* (*Ferrer et al., 2019*; *Miyasaka et al., 2004*; *Ratzinger et al., 2002*; *Zimmerli and Hauser, 2007*), are elevated in dermal LCs and weakly expressed in dermal LC$^{like}$ cells (*Figure 3D*), whereas no substantial differences are detected in expression of *Sirpa, Ccr7, and Cd24a* (*Figure 3E*, *Figure 3—figure supplement 1B*). The reduced expression of *Adgre1* (F4/80) in LC$^{like}$ cells (*Figure 3E*), re-confirms the downregulation of F4/80 surface expression on this cell type observed in our previous flow cytometry analysis (*Figure 1*). Interestingly, both β and γ chains of the IL-2R (*Il2rb* and *Il2rg*), previously reported to be expressed in DCs (*Zelante et al., 2012*), are highly expressed in LC$^{like}$ cells, whereas LCs are the dermal cells expressing the lowest levels among the different DCs and macrophages subpopulations (*Figure 3—figure supplement 1C*).

In summary, the unsupervised clustering of single cells obtained from dermis suggests that LC and LC$^{like}$ cells are two independent cell fractions and distinct from CD207$^+$CD326$^+$ cDC1s as well as from cDC2 CD11b$^+$ and TN DCs, as already shown in conventional flow cytometry analysis (*Figure 1B*).

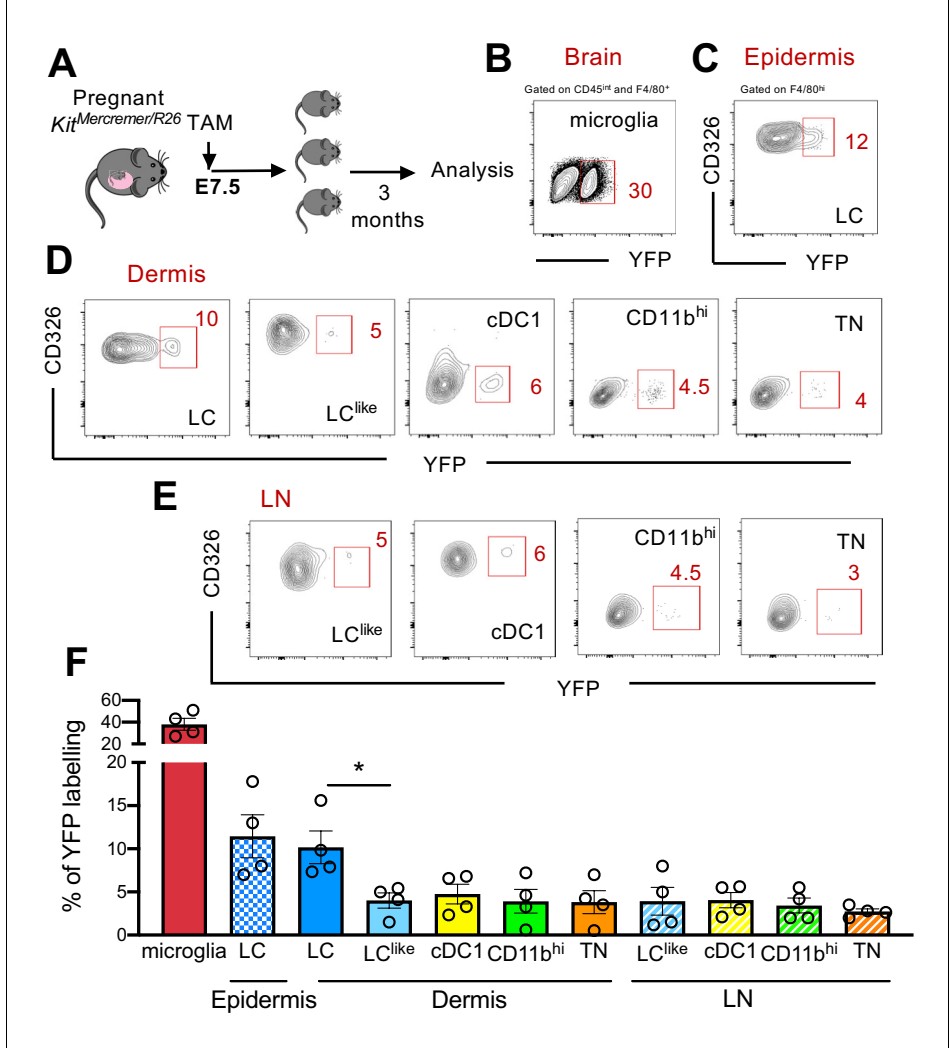

**Figure 4.** Distinct embryonic origin between Langerhans cell (LC) and LC^like cells. (**A**) Single pulse of TAM at E7.5 was given to label Kit^MercreMer/R26 embryos and the percentages of labelled brain microglia (positive control, gated on CD45^intF4/80^hi), epidermal LCs, and dermal LC/dendritic cell (DC) subpopulations were measured at 3 months of age. (**B–D**) Flow cytometry analysis of YFP labelling of microglia (**B**), and each LC and DC subpopulation in the epidermis (**C**), dermis (**D**), and lymph node (LN) (**E**) in Kit^MerCreMer/R26 fate mapping mice. Representative contour plots are shown. (**F**) The mean percentage of YFP^+ cells of brain microglia, epidermal LC, and dermal DC subpopulations (LC, cDC1, LC^like, CD11b^hi, and triple negative [TN] cells). The error bars represent the SEM (n = 4 samples of two to three pooled mice for epidermis/dermis and n = 5 mice for LN). Data from two independent experiments. *p<0.05; two-way ANOVA followed by Bonferroni test. For clarity, non-significant values are not shown.

The online version of this article includes the following source data for figure 4:

**Source data 1.** Percentage of YFP+ cells of brain microglia, epidermal LC,dermal and LN DC subpopulations (LC, cDC1, LClike, CD11bhi,and TN cells).

## Early yolk sac precursors contribute to the development of LC but not LC^like cells

Fate mapping experiments have shown that epidermal LCs derived partially from primitive yolk sac progenitors (*Hoeffel et al., 2012*; *Sheng et al., 2015*); therefore, the developmental origin of LCs is distinct from conventional DCs and resembled more microglia. To study in detail a possible yolk sac origin of distinct cutaneous LC and DC subpopulations, a single injection of tamoxifen (TAM) was

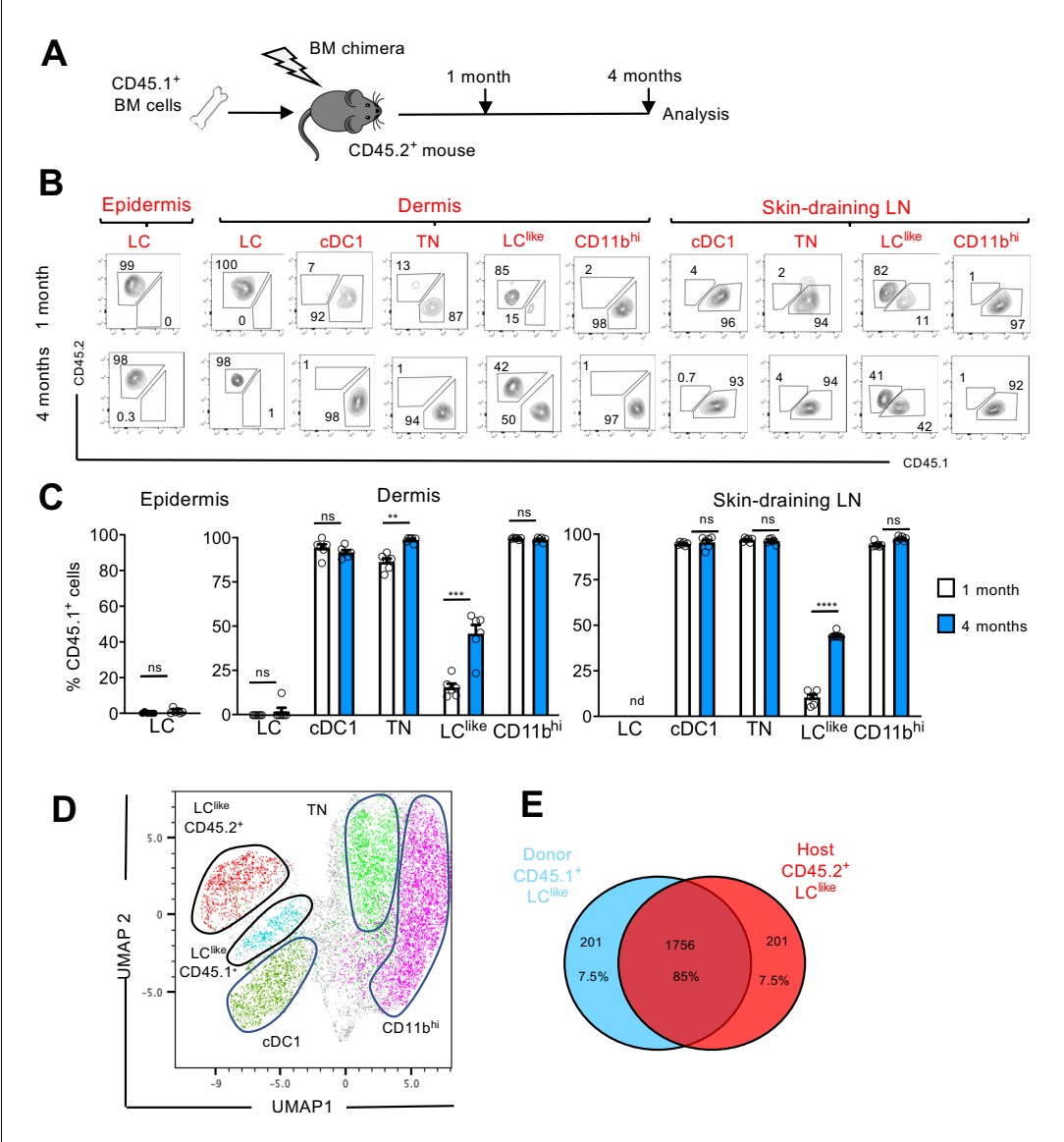

**Figure 5.** LC[like] cells derived from embryonic and adult haematopoiesis have a similar transcriptomic signature. (A) Generation of BM chimeras: CD45.1[+] WT BM cells (10[6]) were transferred into lethally irradiated CD45.2[+] recipient mice. The epidermis, dermis, and draining lymph nodes (LNs) obtained from the reconstituted chimeras were analysed 1 and 4 months later by flow cytometry. (B) Flow cytometry analysis of donor (CD45.1[+]) and host (CD45.2[+]) chimerism in different epidermal, dermal, and skin-draining LN LC and dendritic cell (DC) subpopulations, 1 and 4 months after reconstitution. LC, cDC1, triple negative (TN), LC[like], and CD11b[hi] subsets were gated and analysed for CD45.1 (x-axis) and CD45.2 (y-axis) expression. (C) The percentage of CD45.1 donor cells detected in the epidermis, dermis, and skin-draining LNs of chimeras, 1 or 4 months after reconstitution. Data are represented as mean ± SEM; n = 6 single mice; **p<0.01; ***p<0.001; ****p<0.0001; ns, non-significant; two-tailed Student's t-test. (D) Uniform manifold approximation and projection (UMAP) analysis of distinct LN DC subpopulations obtained from chimeras 4 months after reconstitution, based on the expression of different markers (CD11c, MHCII, CD103, CD11b, CD326, CD207, CD45.1, CD45.2). (E) Transcriptome analysis of LN CD45.1[+] LC[like] cells (n = 3) and LN CD45.2[+] LC[like] (n = 3) cells collected from 10 mice. The Venn diagram shows the percentage of overlapping genes expressed by CD45.1[+] and CD45.2[+] LC[like] cells.

The online version of this article includes the following source data for figure 5:

**Source data 1.** Percentage of CD45.1+ donor cells detected in the epidermis, dermis and skin-draining LNs of mouse chimeras, 1 or4monthsafter reconstitution.

given to E7.5 pregnant *Kit*[MerCreMer]/R26 mice (*Figure 4A*). Three months later, the epidermis, dermis, and brain (microglia as positive control) were collected and isolated cells were then analysed for YFP expression. As previously reported, microglia, the prototype yolk sac-derived macrophage,

were strongly labelled (~40%) (*Figure 4B and F*). However, about 12% of epidermal LCs were YFP labelled, confirming their partial yolk sac origin (*Figure 4C and F*). In comparison, the dermal LC counterparts showed a similar labelling profile (~10%), whereas the remaining dermal DC subpopulations (LC^like, cDC1, CD11b^hi, and TN) showed a significantly lower 5% YFP signal, very likely, attributed to small spillover of labelling in the HSCs (haematopoietic stem cells) (*Figure 4D and F*). Therefore, YS only contributed to LCs but not to dermal LC^like cells. Low YFP labelling was also obtained in all migratory LN DC counterparts (*Figure 4E and F*).

## LC^like DCs derive from both embryonic and adult BM haematopoiesis

LCs are the only cell type from the DC family that originate from self-renewing *radio-resistant embryonic* precursors (*Merad et al., 2002*); other DC subpopulations are short-lived and constantly replenished by BM progenitors (*Kissenpfennig et al., 2005*). To delineate the radio-resistant properties of the newly identified LC^like cells, we generated BM chimeric mice by transplanting congenic CD45.1^+ mouse BM cells into irradiated CD45.2^+ recipients (*Figure 5A*). We then analysed the CD45.1^+/CD45.2^+ ratio in different skin-related DC subpopulations 1 or 4 months after reconstitution.

In the epidermis and dermis, LCs were mostly CD45.2^+, and thus retained their host origins due to local self-renewal (*Figure 5B,C*). By contrast, dermal cDC1, TN, and CD11b^hi DCs exhibited a wholly CD45.1^+ phenotype after just 1 month following reconstitution; this finding means that they are fully BM-derived. Only LC^like cells showed a mixed contribution from both CD45.2^+ host and CD45.1^+ donor cells. In fact, after 1 month following reconstitution, only a minority (~10%) of LC^like cells were replenished by CD45.1^+ cells; this percentage increased to ~50% by 4 months after reconstitution (*Figure 5B,C*).

In skin-draining LNs, we found that cDC1, TN, and CD11b^hi cells were mostly derived from donor CD45.1^+ BM cells, excluding their origins from the radio-resistant LC population. Comparable to its dermal counterpart, only the LC^like cell fraction was split into donor CD45.1^+ and host CD45.2^+ cells, respectively (*Figure 5B,C*). In addition, the contribution of CD45.1^+ donor cells increased over time, from ~10% after 1 month to ~45% after 4 months. This unique temporal replacement suggests a dual origin for LC^like cells, distinguishing this DC fraction from both conventional long-lived radio-resistant self-renewing LCs and short-lived BM-derived DCs.

To allow high-resolution and unbiased data-driven dissection of skin DC subpopulations in the reconstituted chimeric mice, we performed a UMAP analysis of flow cytometry data. Both CD45.1^+ and CD45.2^+ LC^like cells were clearly visible and clustered separately, but in close proximity (*Figure 5D*). Using this dimensional reduction algorithm, we detected that CD11c^+MHCII^hi dermal DC subpopulations could be grouped into five separate clusters: cDC1, TN, CD11b^hi, and two LC^like cell clusters (BM-derived CD45.1^+ and resident CD45.2^+). To investigate the molecular relationship between the resident LC^like cell population and the BM-derived LC^like cells, we performed RNA-seq on LN LC^like cells isolated from chimeric mice (CD45.1^+ donor BM cells into CD45.2^+ recipient mice). Unsupervised hierarchical clustering (Euclidean distance, complete linkage) and principal component analysis (PCA) analysis revealed that both CD45.1^+ and CD45.2^+ LC^like cells clustered closely together (not shown), with ~85% of their gene expression overlapping (*Figure 5E*). The high level of similarity between resident and BM-derived LC^like fractions indicates that the microenvironment, and not the cellular origin, seems to determine the LC^like cell identity.

## LC^like cells display slow turnover kinetics

BM chimeras require full body irradiation, which can damage the local skin microenvironment and attract BM-derived newcomers. This irradiation could, therefore, complicate the analysis of skin-resident cell homeostatic turnover kinetics. To circumvent this issue, we performed a fate mapping study under steady-state conditions using *Kit*^MerCreMer/R26 fate mapping mice. These mice allow for the turnover rates of cell populations derived from BM precursors to be estimated (*Sheng et al., 2015*). We performed our analyses at different time points (1, 4, and 8 months) after TAM injection to ensure a sufficiently long time frame to monitor populations that turn over slowly (*Figure 6A*).

In the epidermis, LCs showed minimal YFP labelling over the entire 8-month chase period; this finding was expected as these cells are not replaced by BM-derived cells (*Figure 6B and C*, left panel). Similarly in the dermis, CD11b^hiF4/80^hiCD326^+CD207^+ cells showed minimal labelling from 1

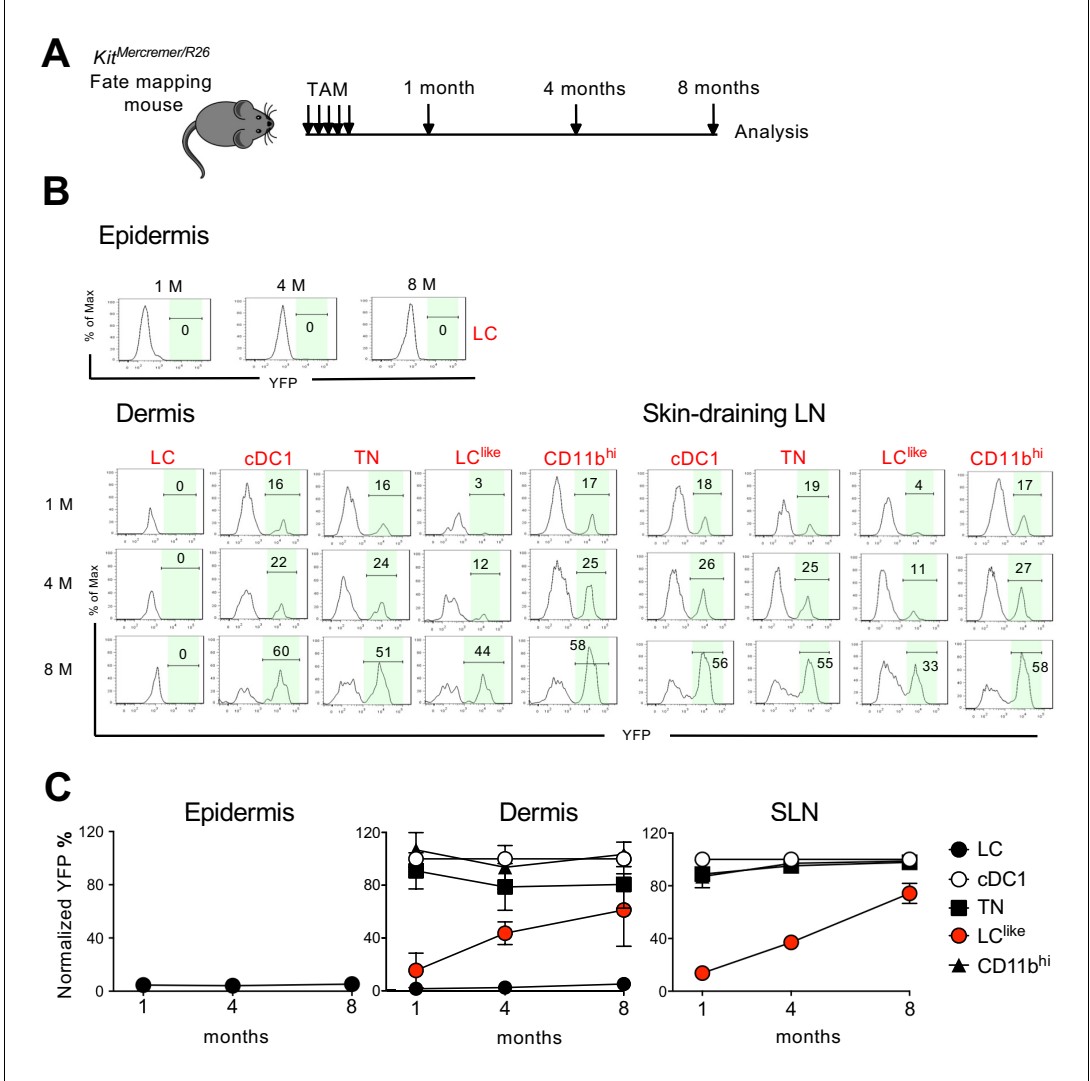

**Figure 6.** Slow turnover kinetics for dermal and lymph node (LN) LC[like] cells. (**A**) Kit[MerCreMer]/R26 mice aged 6 weeks old were injected with tamoxifen five times and groups of six animals were sacrificed 1, 4, and 8 months later for fate mapping analysis. (**B**) Flow cytometry analysis of YFP labelling of each LC and dendritic cell (DC) subpopulation in the epidermis, dermis, and skin-draining LNs in Kit[MerCreMer]/R26 fate mapping mice. Representative histograms are shown. (**C**) The mean percentage of YFP+ cells after normalization to cDC1. Epidermis (left), dermis (middle), and skin-draining LNs (right) were analysed. The error bars represent the SEM (n = 6 mice).

The online version of this article includes the following source data and figure supplement(s) for figure 6:

**Source data 1.** Percentage of YFP+ epidermal, dermal and LN cells analysed at 1, 4 and 8 months after tamoxifen injection.

**Figure supplement 1.** Bar plots showing distinct dermal myeloid cell subset distributions across cell cycle phases.

to 8 months (*Figure 6B and C*, middle panel). We propose that this fraction most likely represents immigrant LCs in the dermis. cDC1, TN, and CD11b[hi] DCs, however, were fully labelled with YFP after just 1 month and the labelling was maintained for the remaining 8 months. This finding is consistent with the fast turnover rate identified for these three DC subsets. By contrast, LC[like] cells gradually accumulated the label from 10% to 60% over the 8-month chase period, supporting that dermis-resident LC[like] DCs are replaced slowly by BM progenitors. In the skin-draining LNs, all DC subsets behaved similarly to their dermal counterparts (*Figure 6B and C*, right panel). Briefly, cDC1, TN, and CD11b[hi] DCs showed a fast turnover by reaching plateau level of labelling after 1 month while LC[like] cells demonstrated a slow turnover rate over the 8-month chase period.

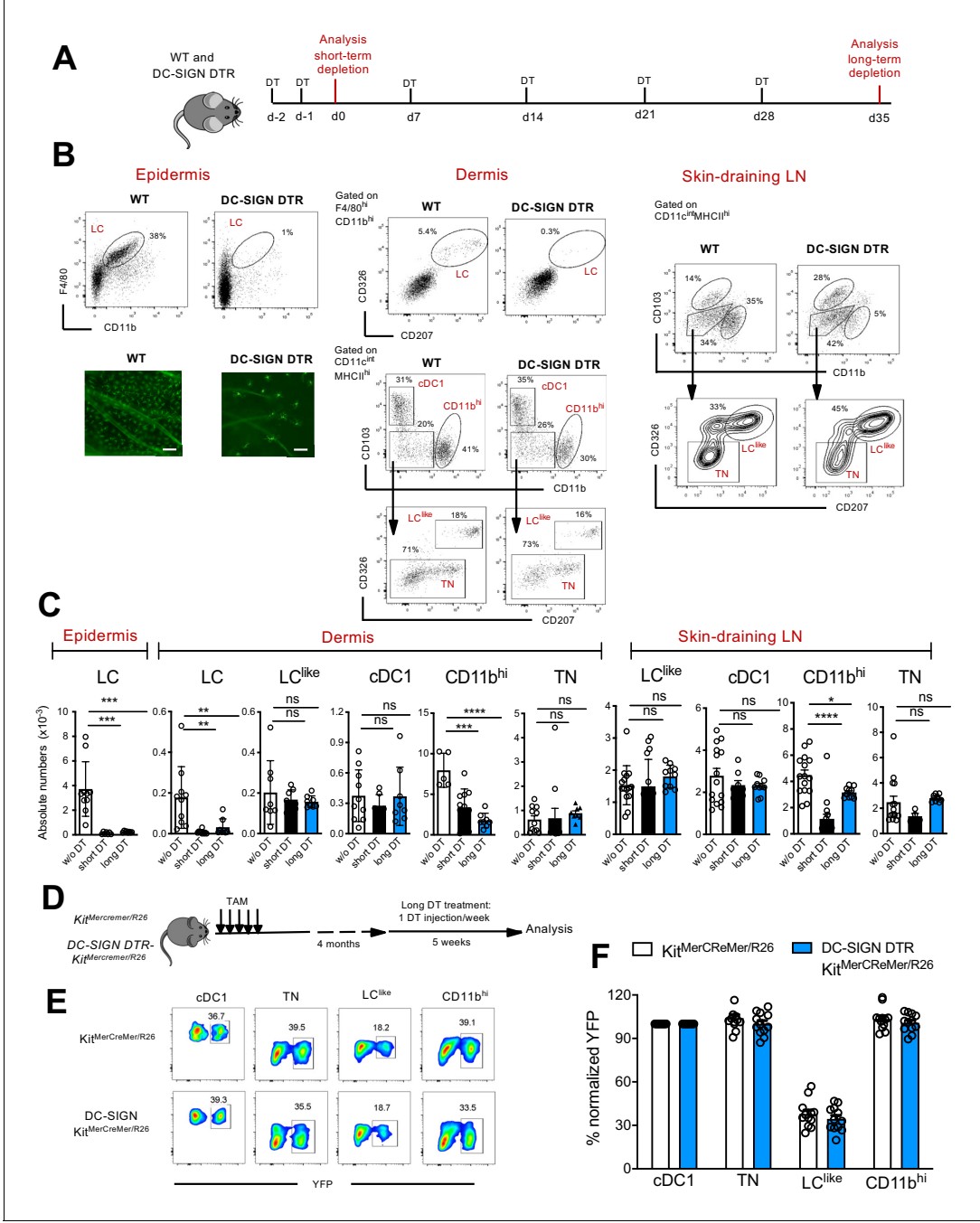

**Figure 7.** Classical Langerhans cells (LCs), but not LC[like] cells, are ablated in vivo in DC-SIGN DTR mice. (**A**) The short-term and long-term depletion protocol in DC-SIGN-DTR mice. (**B**) Representative flow cytometry dot plots of single-cell suspensions from the epidermis (left), dermis (middle), and skin-draining lymph nodes (LNs) (right) obtained from diphtheria toxin (DT)-injected WT and DC-SIGN-DTR mice. All mice were injected (i.p.) with 10 ng/g DT on days −2 and −1 and analysed on day 0. The gating strategy shown in *Figure 1* was followed. Epidermal sheets obtained from WT and DC-SIGN mice were stained for MHC class II (green fluorescence) and analysed by immunofluorescence microscopy (lower left panel). (**C**) The absolute numbers of each gated myeloid cell subset (LC, cDC1, triple negative [TN], LC[like], and CD11b[hi] cells) obtained from the epidermis, dermis, and skin-draining LNs of DT-injected WT and DC-SIGN DTR mice. Data are represented as mean ± SEM; *n* = 8–10 single mice. ****p<0.0001; ***p<0.001; **p<0.01; ns, non-significant; two-way ANOVA statistical test Bonferroni test. (**D**) Fate mapping analysis in DC-SIGN DTR-Kit[MerCreMer]/R26 mice. Mice aged 6 weeks old were orally gavaged with TAM. After 4 months, DT was injected i.p. weekly for 5 weeks to ensure long-term LC depletion. (**E**) Representative contour plots showing the YFP labelling of distinct LN dendritic cell (DC) subpopulations in DT-treated Kit[MerCreMer]/R26 and DC-SIGN DTR-Kit[MerCreMer]/R26 mice. (**F**) The percentage of normalized YFP labelling detected in DC subpopulations (LC, cDC1, TN, LC[like], and CD11b[hi] cells) of the skin-draining LNs. Normalization was performed as described in *Figure 6*; data are represented as mean ± SEM; *n* = 12 single mice.

*Figure 7 continued on next page*

*Figure 7 continued*

The online version of this article includes the following source data and figure supplement(s) for figure 7:

**Source data 1.** Absolute numbers of epidermal, dermal and LN DC subpopulations after short and long term depletion in DC-SIGN-DTR mice.

**Figure supplement 1.** Epidermal LCs isolated from DC-SIGN DTR mice express human HB-EGF.

Cell cycle analysis was performed for dermal LC, LC[like], and different DC subsets based on scRNA-seq (*Figure 6—figure supplement 1*). The LC[like] subset exhibited higher proliferating capability than LC and other DC subsets, consistent with the previous findings that LCs are dividing extremely slow (*Ginhoux and Merad, 2010*) and conventional DCs do not proliferate in tissues and mainly depend on their BM progenitors for expansion (*Liu and Nussenzweig, 2010*). Overall, we showed that LC[like] cells displayed slower turnover kinetics than other DC subpopulations and higher proliferating capability to refill the emigration gap.

## LC[like] cells are not derived from classical LCs

To interrogate the relationship between LC and LC[like] cells, we exploited a novel DC-SIGN-DTR transgenic mouse strain (*Figure 7—figure supplement 1A*), which allowed us to deplete epidermal and dermal LCs without affecting the LC[like] cell pool (*Figure 7*). Although not detectable by flow cytometry on the cell surface (*Figure 1—figure supplement 1B*), we measured DC-SIGN (or CD209a) specific mRNA levels in murine LCs as well as in CD11b[hi] DCs, the latter already known to express this receptor (*Cheong et al., 2010*), whereas cDC1, TN DCs, or LC[like] cells were negative (*Figure 7—figure supplement 1B*), a result which was corroborated by the scRNA analysis of dermal cells (*Figure 2*). Since DC-SIGN expression has never been reported for LCs, quantitative PCR (qPCR) analysis was performed to detect the expression of DTR (known as human heparin-binding EGF-like growth factor [HBEGF]) in LCs obtained from DC-SIGN-DTR mice. Accordingly, *HBEGF* mRNA was detected only in LC isolated from epidermis of DC-SIGN DTR mice and was absent in LCs obtained from WT mice (*Figure 7—figure supplement 1C*). Thus, the decrease in LCs observed after diphteria toxin (DT) injection was achieved due to the high sensitivity of the DT-DTR system (*Ruedl and Jung, 2018*), although no DC-SIGN was measurable on the cell surface of LCs from DC-SIGN DTR[+] mice. To exclude a downregulation of the DC-SIGN receptor in LCs upon maturation, we sorted epidermal LCs, cultured them overnight with GM-CSF and LPS and compared by qPCR the *Cd209a* expression between unstimulated and stimulated epidermal LC fractions. Clearly no *Cd209a* downregulation was observed in activated LCs (*Figure 7—figure supplement 1C*); therefore, an eventual transition from maturing dermal DC-SIGN[pos] LCs to dermal DC-SIGN[neg] LC[like] cells can be excluded.

We established short and long depletion protocols (*Figure 7A*) to capture even potentially very slowly migrating 'LC derivatives' (*Bursch et al., 2007*). In the DT-treated DC-SIGN DTR mice, LCs were efficiently depleted in both the epidermis and dermis by the short-term and long-term depletion protocols (*Figure 7A–C*). We also found that cells in the CD11b[hi] cell fraction were affected by the DT treatment; this was particularly evident during the short-term depletion protocol, in which the cell numbers were reduced by ~80% (*Figure 7C*). Importantly, cDC1, TN, and LC[like] cell numbers were unaffected and thus were comparable between DT-injected WT and DC-SIGN mouse strains. These results strongly support the independency of LC[like] cells from classical bona fide epidermal LCs.

To further confirm that LC[like] cells represent a distinct cell lineage from LCs, we crossed DC-SIGN DTR mice with a *Kit*[MerCreMer]/R26 fate mapping mouse, which would enable us to trace BM-derived cells in absence of LC. We treated these mice (DC-SIGN DTR-*Kit*[MerCreMer]/R26) with TAM and then injected them with DT for 5 weeks to maintain long-term LC depletion (*Figure 7D*). Although epidermal LCs were absent over the whole period, the YFP labelling profiles of skin-derived LN DC subsets, including the LC[like] fraction, were comparable between DT-injected DC-SIGN DTR[+]-*Kit*[MerCreMer]/R26 and DC-SIGN DTR[neg]-*Kit*[MerCreMer]/R26 mice (*Figure 7E,F*). These data show that in absence of LC, the replenishment of LC[like] cells by BM-derived cells is not affected.

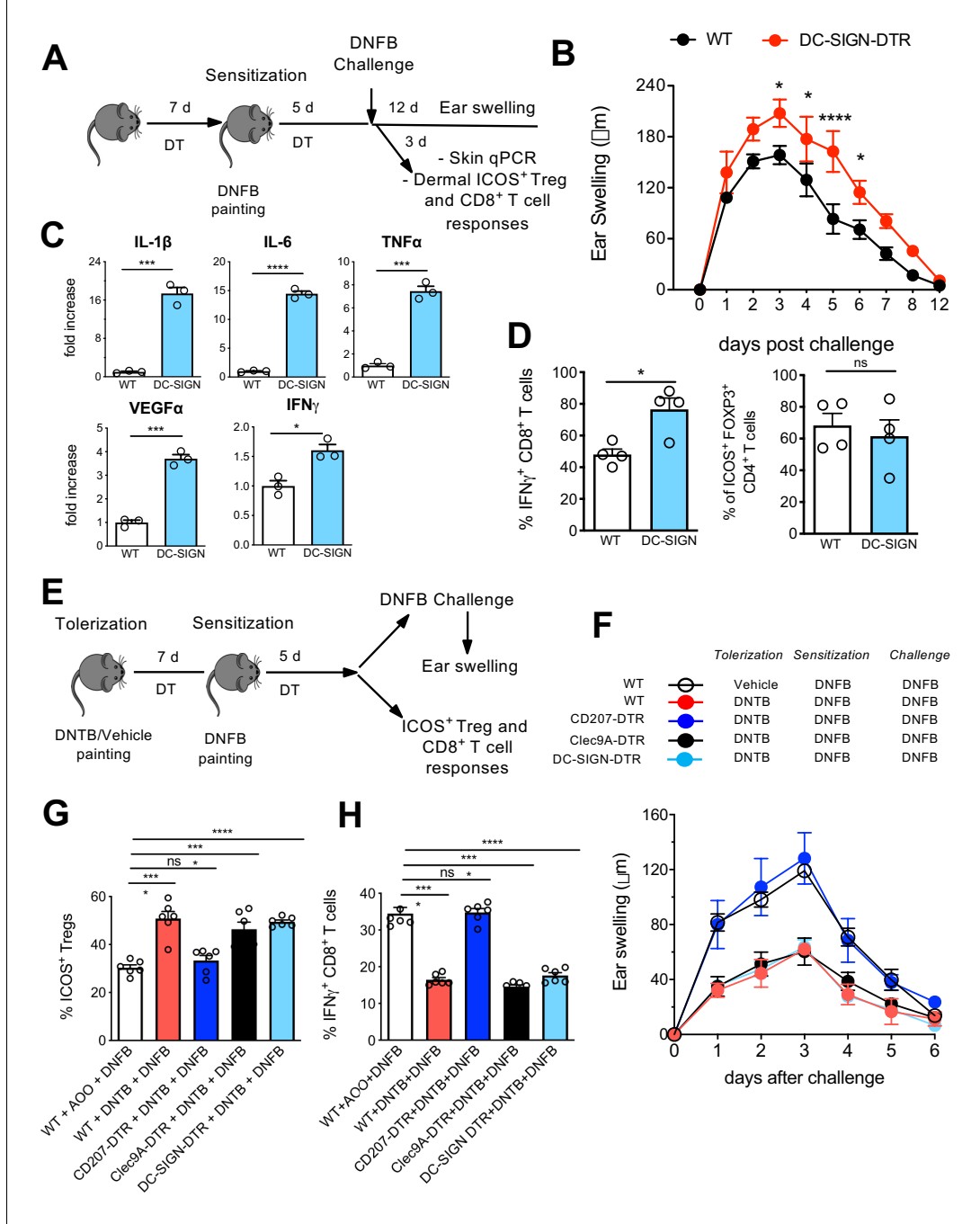

**Figure 8.** Differential contribution of Langerhans cell (LC) and LC[like] in skin immune responses. (**A**) Diphteria toxin (DT)-treated WT and DC-SIGN DTR mice were sensitized with 0.5% 2,4-dinitrofluorobenzene (DNFB) (applied to the shaved back skin) and ear-challenged 5 days later with 0.1% DNFB. (**B**) Ear swelling response of challenged WT and DC-SIGN DTR mice was determined over a 12-day period post challenge. (**C**) Quantitative PCR analysis of distinct inflammatory cytokines and growth factors in ears collected at day 3 post challenge. The error bars represent the SEM (n = 3 mice). *p<0.05; ***p<0.001; ****p<0.0001; two-tailed Student's *t*-test. (**D**) Percentages of dermal IFN-γ-producing CD8+ T-cells and dermal CD4+Foxp3+ICOS+ regulatory T-cells at day 3 post challenge. Single-cell suspensions were generated from the dermis and the cells were re-stimulated (5 hr) with PMA/ Ionomycin to detect IFN-γ production or directly stained for determination of activated T$_{regs}$. The error bars represent the SEM (n = 4 mice). *p<0.05; ns, non-significant; two-tailed Student's *t*-test. Gating strategy is shown in *Figure 8—figure supplement 1*. (**E**) DT-treated WT, CD207-DTR, Cleac9A-DTR, and DC-SIGN DTR mice were tolerized with 1% 2,4-dinitrothiocyanobenzene hapten (DNTB) (applied to the shaved abdomen skin). After 7 days, the mice were sensitized with 0.5% DNFB (applied to the shaved back skin) and ear-challenged 5 days later with 0.1% DNFB. (**F**) The ear swelling response of painted mice was determined over a 6-day period post challenge. (**G**) Percentage of activated CD4+Foxp3+ICOS+ regulatory T-cells in the draining lymph nodes (LNs) of mice, 5 days after vehicle or DNTB painting. The data were pre-gated on singlet, live CD3+CD4+Foxp3+ T$_{reg}$ cells. The

*Figure 8 continued on next page*

Figure 8 continued

error bars represent the SEM (n = 6 mice). ****p<0.0001; ns, non-significant; two-way ANOVA. (**H**) Percentages of IFN-γ-producing CD8$^+$ T-cells in the draining LNs of vehicle or DNTB-painted mice followed by DNFB sensitization. Single-cell suspensions were generated from the LNs and the cells were re-stimulated (5 hr) with PMA/Ionomycin to detect IFN-γ production. The data were pre-gated on singlet, live CD3$^+$ CD8$^+$ T-cells. The error bars represent the SEM (n = 6 mice). ****p<0.0001; ns, non-significant; two-way ANOVA.

The online version of this article includes the following source data and figure supplement(s) for figure 8:

**Source data 1.** Percentages of IFNgamma+ CD8+ T cells and ICOS+ FOXP3+ T regs measured in distinct skin immune repsonses.

**Figure supplement 1.** Gating strategy with representative flow cytometry samples.

**Figure supplement 2.** Transgenic DTR mouse strains used for in vivo abaltion of CD207+ and Clec9A+ cells.

## LCs suppress cutaneous CHS responses within the skin

We next investigated the specific local contribution of LCs during CHS. Therefore, DT-injected WT and DC-SIGN DTR mice were sensitized with 0.5% 2,4-dinitrofluorobenzene (DNFB) and challenged at day 5 with 0.2% DNFB to induce a CHS reaction. The ear swelling was subsequently monitored over 12 days. In another group of mice, ears were processed 3 days post challenge for qPCR analysis as well as processed for cell isolation (*Figure 8A*). In absence of LCs, clearly a pronounced increased ear swelling persisted over 10 days (*Figure 8B*), a phenotype which was already reported in other LC-ablating transgenic mouse strains such as human langerin-DTA (hu-DTA) and human langerin-DTR (huDTR) mice (*Kaplan et al., 2005*; *Bobr et al., 2010*). Proinflammatory cytokines such as IL-6, TNF-α, and IL-1β were clearly significantly upregulated in ears of DT-treated DC-SIGN DTR mice (*Figure 8C*, upper panel). VEGFα, a biomarker for keratinocyte impairment (*Canavese et al., 2010*, *Bae et al., 2015*), was also clearly augmented in absence of LCs (*Figure 8C*, lower panel). In addition, higher IFN-γ levels were measured in ears lacking LCs (*Figure 8C*, lower panel), values that correlated with an augmented CD8$^+$ T-cell-dependent IFN-γ response observed by flow cytometry analysis (*Figure 8D*, *Figure 8—figure supplement 1*). In accordance with previously published data (*Igyarto et al., 2009*), no major difference was observed in activated dermal ICOS$^+$ T$_{regs}$ in presence or absence of LCs (*Figure 8D*, right panel, *Figure 8—figure supplement 1A*).

## Depletion of LC$^{like}$ cells, but not LCs, breaks tolerance to DNTB

In our final assays, we aimed to determine the contribution of distinct skin-resident DC subpopulations to the induction of tolerance to CHS. Here, we injected a series of different DTR mice, including CD207-DTR, Clec9A-DTR, and DC-SIGN DTR mice, with DT to deplete the target cells over the whole period of tolerization, sensitization, and final challenge (*Figure 8E*). Of note, each DTR mouse strain shows a different LC/DC depletion profile: DT-treated CD207-DTR mice lack all epidermal and dermal CD207 expressing cells (LC, LC$^{like}$, and cDC1) (*Figure 8—figure supplement 2A–C*), Clec9-DTR mice lack cDC1s (*Figure 8—figure supplement 2A,D and E*), and DC-SIGN lack epidermal and dermal LCs and CD207$^-$CD326$^-$CD11b$^{hi}$ DCs (*Figure 7*).

We next established an experimental 2,4-dinitrothiocyanobenzene hapten (DNTB)-mediated tolerance mouse model. Here, we induced immune tolerance against the strong contact sensitizer DNFB by epicutaneous application of an innocuous DNTB. We applied DNTB to the shaved abdomen of DT-injected WT, CD207-DTR, Clec9A-DTR, and DC-SIGN DTR mice 7 days prior to subsequent DNFB sensitization (*Figure 8E*). As a positive control, we established a group of WT mice that was not treated and tolerized with DNTB. When the mice were ear-challenged with DNFB, we found that the non-tolerized WT mice developed robust CHS, as evidence by ear swelling that increased over time upon DNFB challenge. CD207-DTR mice were clearly not tolerized against DNFB, since they showed robust ear inflammation over time. Such ear swelling was not observed in DNTB-tolerized WT, Clec9A-DTR, and DC-SIGN-DTR mice (*Figure 8F*). CHS is conferred by cytotoxic T-cell-mediated skin inflammation induced by exposure to strong contact sensitizers, such as DNFB. We thus analysed activated IFN-γ-producing CD8$^+$ cytotoxic T-cells in the skin-draining LNs by flow cytometry on day 5 post DNFB sensitization. In line with the observed ear swelling profile, we found that a low IFN-γ-producing CD8$^+$ cytotoxic T-cell response was restricted to DNTB-tolerized WT, Clec9A-DTR, and DC-SIGN-DTR mice. This response was comparatively enhanced in CD207-DTR

and WT mice (*Figure 8G*, *Figure 8—figure supplement 2B*). The diminished level of ear inflammation in DNTB-tolerized WT, Clec9A-DTR, and DC-SIGN-DTR mice correlated with an increased presence of activated ICOS⁺CD4⁺FOXP3⁺ T$_{regs}$ in the LN, which exhibit suppressive activity in CHS to DNFB (*Vocanson et al., 2010*). Remarkably, we did not observe this phenotype in DNTB-tolerized CD207-DTR or non-tolerized WT mice (*Figure 8H*, *Figure 8—figure supplement 2B*). Because of this clear difference between CD207-DTR mice (that lack CD207⁺ LCs/LC-like cells and CD207⁺ DC1s) and DC-SIGN DTR mice (that lack only CD207⁺ LCs) and Clec9A-DTR mice (that lack only CD207⁺ cDC1), we conclude that only LC$^{like}$ cells critically contribute to tolerance induction.

## Discussion

Epidermal LCs are the only APCs localized in the epidermis. These cells were recently re-defined as 'macrophages in DC clothing' due to their unique ontogeny, and self-renewing and radio-resistant characteristics (*Doebel et al., 2017*). By contrast, there are multiple DC and macrophage subpopulations that reside in the dermis (*Tamoutounour et al., 2013*). Although these dermal DCs share some common markers with LCs (such as langerin [CD207] and EpCAM [CD326]), they constitute a distinct cell lineage on the basis of their developmental origins and cytokine requirements (*Bursch et al., 2007*; *Ginhoux et al., 2007*; *Poulin et al., 2007*). Three CD207⁺ DC subpopulations have been described in the skin-draining LN: two subpopulations are skin-derived and one subpopulation originates from the BM (*Bursch et al., 2007*; *Douillard et al., 2005*; *Henri et al., 2001*; *Romani et al., 2010*). Due to this diverse skin-resident DC network, it became evident that not only LCs but other skin-derived DCs might be involved either in tolerance or immune response induction in draining LNs.

Although it is commonly believed that the journey of an LC starts from the epidermis and ends in the skin-draining LN after a transit through the dermis in steady state, we found that it is in fact their look-alike counterparts, LC$^{like}$ cells, that migrate to the draining LNs. Our new insight was gained by re-analysing established mouse strains (*Kit$^{MerCreMer}$/R26* mice) and exploiting newly generated transgenic mouse strains (DC-SIGN-DTR mice and DC-SIGN-DTR- Kit$^{MerCreMer}$/R26 fate mapping mice), which allowed us to visualize, with increasing resolution, the in vivo dynamics of skin-resident DCs under steady state. We first characterized and re-defined different DC/LC subsets in the dermis by flow cytometry and scRNA-seq analysis, which delineated classical F4/80$^{hi}$ LC and four different DC subsets, namely cDC1, TN DCs, CD11b$^{hi}$ DCs, and an unappreciated CD11b$^{low}$F4/80$^{low}$ LC$^{like}$ cell fraction. With the exception of classical F4/80$^{hi}$ LCs, we found all of these cells in the migratory CD11c$^{int}$MHCII$^{hi}$ DC fraction of the skin-draining LN. This finding suggests that the majority of migratory CD326⁺CD207⁺ DCs are CD103⁺ cDC1 and CD103⁻ LC$^{like}$ cells and not classical CD11b$^{hi}$F4/80$^{hi}$ LCs which are hardly seen in the LN if not at all. Corroborating evidence for differential migratory behaviours among different skin DCs was provided by real-time intravital two-photon microscopy. Under steady-state conditions, due to the structural integrity of the basement membrane, epidermal LCs are sessile with static and almost immobile dendrites. In contrast, dermal DC subpopulations are actively crawling through the dermal interstitial space at high velocity even in absence of inflammation suggesting that continuous migration to LN is a steady-state property of dermal DCs and not epidermal LCs (*Ng et al., 2008*; *Shklovskaya et al., 2011*).

Our analysis of dermal DCs is in full agreement with recent published data by *Henri et al., 2010*. The DC family in the dermis was likewise disentangled in five subpopulations: two subsets lacking the expression of CD207 (CD207⁻CD11b⁻ [TN] and CD207⁻CD11b⁺ [CD11bhi]) and three expressing CD207 (CD11b$^{int}$CD207$^{++}$ mLCs, CD11b$^{low/-}$CD207⁺CD103⁺ [cDC1], and CD11b$^{low}$CD207⁺CD103⁻ [LC$^{like}$]) (*Henri et al., 2010*).

Similarly, cutaneous LNs were distinguished in five analogous subpopulations including mLCs, which were defined for their characteristics in radioresistance and not for the expression of classical LC markers (CD11b$^{hi}$ and F4/80$^{hi}$). Henri et al. speculated that LN LCs downregulated CD11b and F4/80 expression (*Henri et al., 2010*) and therefore these markers lost their discriminatory power to segregate distinct CD207⁺ cells in the LN.

To circumvent the 'complication' of the potential shift in phenotype, we adopted an alternative approach based on genetic fate mapping analyses which allowed to trace cell lineages between distinct LC and DC subpopulations avoiding lethal irradiation and generation of chimeric mice. First, our E7.5 embryo 'labelling strategy' demonstrated that only LCs are partially yolk sac-derived

(*Sheng et al., 2015*), but not the other migratory DCs' subpopulations, inclusive of LC[like] cells. Second, our detailed analyses of the fate mapping kinetics revealed that the radio-resistant and radio-sensitive CD11b[low]CD207[+]CD103[-] subpopulations described by Henri et al. represented instead a truly homogeneous radio-resistant LC[like] subpopulation, which is gradually replaced over time by BM-derived progenitors. Furthermore, we corroborated their 'LC independency', since long-term absence of LCs did not affect the numbers of LC[like] cells in our DC-SIGN DTR mouse model. We also ruled out a possible downregulation of DC-SIGN during LC maturation which excludes the transition of a dermal DC-SIGN[pos] LC to a DC-SIGN[neg] LC[like] cell. Accordingly, our analysis delineated only four, and not five, LN migratory DC subpopulations, excluding LCs. The phenotypes, transcription profiles, and cytokine requirements of dermal cDC1, TN, and CD11b[hi] DCs have been extensively described (reviewed in *Clausen and Stoitzner, 2015*); however, there has been comparatively less attention given to the LC[like] subpopulation.

Unsupervised clustering of scRNA-seq trascriptome data of dermal cells indicated that LC and LC[like] cells, although sharing some common myeloid cell markers and TFs, are two independent cell fractions and clearly distinct from macrophages and the other skin DC subpopulations. However, unlike LCs, which are BM-independent, radio-resistant, and self-renewing (*Ghigo et al., 2013*; *Hoeffel et al., 2012*), LC[like] cells represent a radio-resistant population that is progressively replaced postnatally by BM-derived precursors. Similar to resident macrophages in tissues, such as skin, gut, kidney, and heart, LC[like] cells have a dual origin involving both embryonic (but not yolk sack-like LCs) and adult haematopoiesis (*Molawi et al., 2014*; *Sheng et al., 2015*; *Soncin et al., 2018*). Unlike other skin DC subpopulations, which are short-lived and exhibit a high turnover rate, we show that fetal-derived LC[like] cells are long-lived, show higher proliferation rates than conventional DCs, and are replaced very slowly by BM-derived cells. These fetal-derived and BM-derived LC[like] cells co-exist together in adult tissue, and although derived from different origins, they show high similarity. This finding suggests that it is the local tissue microenvironment and not the cellular origin that shapes their final identity. The existence of an LC-independent radio-resistant dermal DC fraction was previously observed in other study that described the presence of an in situ proliferating, radio-resistant dermal DC subpopulation not only in the murine but also in human dermis (*Bogunovic et al., 2006*). Similarly, a langerin-expressing dermal myeloid CD1[+] DC fraction unrelated to XCR1[+] DCs and LCs has been reported in human dermis (*Bigley et al., 2015*). It is likely that these cells are the LC[like] cells described here and further studies will be needed to analyse the potential relationship between murine CD207[+] LC[like] cells and human myeloid CD207[+] DC counterparts.

Although all dermal DCs migrate into LNs in a CCR7-dependent fashion (*Förster et al., 1999*), LC[like] cells seem to migrate at slower rate than other DCs under steady-state conditions. Similar slow trafficking dynamics was originally attributed to LCs (*Bursch et al., 2007*; *Ruedl et al., 2000*) but we now strongly believe that these previously reported slow migratory cells are in fact LC[like] cells.

To further rule out the possibility that epidermal F4/80[hi]CD11b[hi] LC downregulate CD11b and F4/80 and turn into F4/80[low]CD11b[low] LC[like] cells in the dermis, we exploited a novel DC-SIGN DTR transgenic mouse strain where LCs, but not LC[like] cells, could be ablated. Even long-term depletion (6 weeks) of epidermal and dermal LCs had no effect on the numbers of LC[like] cells in the dermis and LNs while maintaining their LC[like] YFP-labelling profile in the absence of epidermal LCs in DC-SIGN DTR-*Kit*[MerCreMer]/R26 mice.

The contribution of LCs to hapten sensitization is still controversial and matter of debate since the phenotype observed in different LC depleting mouse models is ranging from reduced to exaggerated CHS reactivity (*Kaplan et al., 2005*; *Bennett et al., 2005*; *Noordegraaf et al., 2010*, *Bobr et al., 2010*; *Clausen and Stoitzner, 2015*). The depletion of LCs, in the herein presented novel DC-SIGN DTR transgenic mouse line, resulted in enhanced CHS, a similar phenotype also reported in hu-DTA and huDTR mice (*Kaplan et al., 2005*; *Bobr et al., 2010*). The T$_{regs}$-independent LC-mediated suppressive effect is restricted locally to the skin since several proinflammatory cytokines (IL-6, TNF-$\alpha$, and IL-1$\beta$) as well as effector IFN-$\gamma$ CD8[+] T-cells are enhanced at the challenged cutaneous side in absence of LCs. Interestingly VEGF-$\alpha$, recently reported as a novel biomarker for pathological cutaneous alterations such as psoriasis (*Canavese et al., 2010*, *Bae et al., 2015*), is highly upregulated in LC-deficient mice which indicates an involvement of LCs in controlling keratinocytes in their VEGF production. In summary, our data corroborate the view that LCs data

corroborate the view that LCs are one of the main immune regulators within the skin (*West and Bennett, 2017*).

The role of different skin-resident DCs in triggering tolerance is also widely debated. Several studies have demonstrated that LCs are crucial for inducing tolerance (*Igyarto et al., 2009*; *Kautz-Neu et al., 2011*; *King et al., 2015*; *Price et al., 2015*; *Shklovskaya et al., 2011*) by stimulating T$_{regs}$ and anergizing CD8$^+$ T-cells. This effect has been shown, for example, in an experimental CHS model (*Gomez de Agüero et al., 2012*). Conversely, others have demonstrated the specific contribution of dermal DCs into maintaining skin tolerance (*Azukizawa et al., 2011*; *Tordesillas et al., 2018*). Given that we only detected LC$^{like}$ cells and not bona fide LCs in the draining LNs, we readdressed this issue in a cutaneous tolerance model towards the weak contact allergen, DNTB. We used our novel inducible DC-SIGN DTR mice, in which LCs but not dermal or LN LC$^{like}$ cells can be ablated, together with other CD207$^+$CD326$^+$ LC/DCs depleting DTR strains (inducible CD207-DTR and Clec9A-DTR mice). Deletion of LCs or cDC1 had no effect on tolerance induction, while deletion of both LCs and LC$^{like}$ cells prevented tolerance induction. This finding demonstrates that migratory LC$^{like}$ cells, which are capable of reaching the draining LNs, are responsible for the expansion of ICOS$^+$CD4$^+$FOXP3$^+$ T$_{regs}$ and the suppression of a cytotoxic T-cell response.

In summary, our genetic fate mapping approach, used to delineate the complex skin DC network, does not support the established paradigm of LCs as being the main 'prototype' migrating APCs to draining LNs under homeostatic conditions (*Wilson and Villadangos, 2004*). We propose, rather, that LCs at steady state, similar to other tissue-resident macrophages, are sessile and act locally in the skin, whereas dermal LC$^{like}$ cells assume many of the functions previously attributed to LCs. The identification of this novel migratory dermal LC$^{like}$ subpopulation opens new avenues and approaches in the development of treatments to cure diseases such as contact allergic dermatitis and other inflammatory skin disorders like psoriasis.

# Materials and methods

## Key resources table

| Reagent type (species) or resource | Designation | Source or reference | Identifiers | Additional information |
|---|---|---|---|---|
| Antibody | Rat anti-mouse CD45 (30-F11) | Biolegend | Cat#: 103108; RRID: AB_312972 | FACS (1:600; 100 μl per test) |
| Antibody | Rat anti-mouse CD45 (30-F11) | Biolegend | Cat#: 103114; RRID: AB_312978 | FACS (1:600; 100 μl per test) |
| Antibody | Rat anti-mouse CD11b (M1/70) | Becton Dickinson- BD | Cat#: 565976; RRID:AB_2721166 | FACS (1:600; 100 μl per test) |
| Antibody | Rat anti-mouse F4/80 (BM8) | Biolegend | Cat#: 123114; RRID: AB_893490 | FACS (1:600; 100 μl per test) |
| Antibody | Rat anti-mouse Ly6c (HK1.4) | Biolegend | Cat#: 128036; RRID: AB_2562352 | FACS (1:600; 100 μl per test) |
| Antibody | Rat anti-mouse I-A/I-E antibody (M5/114.15.2) | Biolegend | Cat#: 107632; RRID: AB_10900075 | FACS (1:600; 100 μl per test) |
| Antibody | Hamster anti-mouse CD11c (N418) | Biolegend | Cat#: 117324; RRID: AB_830646 | FACS (1:600; 100 μl per test) |
| Antibody | Hamster anti-mouse CD103 (2E7) | Biolegend | Cat#: 121416; RRID: AB_1574957 | FACS (1:600; 100 μl per test) |
| Antibody | Mouse anti-mouse CD209 (clone: MMD3) | Thermo Fisher Scientific | Cat#: 50-2094-82; RRID:AB_11219065 | FACS (1:600; 100 μl per test) |
| Antibody | Rat anti-mouse CD326 (G8.8) | Biolegend | Cat#: 118231; RRID: AB_2632774 | FACS (1:600; 100 μl per test) |
| Antibody | Mouse anti-mouse CD207 (clone: 4C7) | Biolegend | Cat#: 144204; RRID: AB_2561498 | FACS (1:600; 100 μl per test) |
| Antibody | Mouse anti-mouse CD45.1 (A20) | Biolegend | Cat#: 110726; RRID: AB_893347 | FACS (1:600; 100 μl per test) |

*Continued on next page*

*Continued*

| Reagent type (species) or resource | Designation | Source or reference | Identifiers | Additional information |
|---|---|---|---|---|
| Antibody | Mouse anti-mouse CD45.2 (104) | Biolegend | Cat#: 109830; RRID: AB_1186103 | FACS (1:600; 100 μl per test) |
| Antibody | Rat anti-mouse CD3 (17A2) | Biolegend | Cat#: 100306; RRID: AB_312670 | FACS (1:500; 100 μl per test) |
| Antibody | Rat anti-mouse CD4 (GK1.5) | Biolegend | Cat#: 100414; RRID: AB_312699 | FACS (1:600; 100 μl per test) |
| Antibody | Rat anti-mouse CD8 (53–6.7) | Biolegend | Cat#: 100722; RRID: AB_312761 | FACS (1:600; 100 μl per test) |
| Antibody | Rat anti-mouse FOXP3 (MF-14) | Biolegend | Cat#: 126407; RRID: AB_1089116 | FACS (1:600; 100 μl per test) |
| Antibody | Hamster anti-mouse ICOS (15F9) | Biolegend | Cat#: 107705; RRID: AB_313334 | FACS (1:600; 100 μl per test) |
| Antibody | Rat anti-IFN-gamma (XMG1.2) | Biolegend | Cat#: 505810; RRID:AB_315404 | FACS (1:600; 100 μl per test) |
| Antibody | Fc-R block (2.4G2) | Self-made | N/A | Blocking step (1:100; 1000 ml per sample) |
| Chemical compound, drug | Brefeldin A | Sigma-Aldrich | Cat#: B7651 | 10 μg/ml |
| Chemical compound, drug | Phorbol 12-myristate 13-acetate | Sigma-Aldrich | Cat#: 79346 | 10 μg/ml |
| Chemical compound, drug | Ionomycin | Sigma-Aldrich | Cat#: I0634 | 10 μg/ml |
| Chemical compound, drug | Collagenase D | Roche | Cat#: 11088882001 | 1 mg/ml |
| Chemical compound, drug | Dispase II | Gibco | Cat#: 17105041 | 1 U/ml |
| Chemical compound, drug | Ficoll-Paque | GE Healthcare | Cat#: 17144003 | |
| Chemical compound, drug | Percoll | Merck | Cat#: P4937-500ML | |
| Chemical compound, drug | Diphtheria toxin | Sigma-Aldrich | Cat#: D0564 | 20 ng DT/g body weight, i.p. |
| Chemical compound, drug | Tamoxifen | Sigma-Aldrich | Cat#: T5648 | 4 mg TAM for 5 consecutive days by oral gavage for adult labelling. Pregnant mice (E7.5) were injected once with 16 mg TAM for embryo labelling. |
| Chemical compound, drug | IMDM | Thermo Fisher | Cat#: 12440046 | |
| Chemical compound, drug | Ammonium thiocyanate | Sigma-Aldrich | Cat#: 221988 | |
| Chemical compound, drug | 5,5'-Dithio-bis-2-nitrobenzoic acid (DNTB) | Sigma-Aldrich (Lancaster Synthesis) | Cat#: D8130 | |
| Chemical compound, drug | 1-Fluoro-2,4-dinitrobenzene (DNFB) | Sigma-Aldrich | Cat#: D1529 | |
| Chemical compound, drug | Acetone | Sigma-Aldrich | Cat#: 650501 | |
| Chemical compound, drug | Saponin | Sigma-Aldrich | Cat#: S7900 | |
| Chemical compound, drug | TRIzol reagent | Thermo Fisher Scientific | Cat#: 15596026 | |
| Commercial assay or kit | RNAsimple Total RNA kit | Tiangen Biotech Ltd | Cat#: DP419 | |

*Continued on next page*

*Continued*

| Reagent type (species) or resource | Designation | Source or reference | Identifiers | Additional information |
|---|---|---|---|---|
| Commercial assay or kit | Foxp3 staining buffer | eBioscience | Cat#: 00-5521-00 | |
| Commercial assay or kit | Cytofix/cytoperm | eBioscience | Cat#: 51-2090KZ | |
| Commercial assay or kit | Ovation Universal RNA-seq system | NuGEN Technologies | Cat#: 0343–32 | |
| Commercial assay or kit | DNA High Sensitivity Reagent Kit | Agilent, Santa Clara, CA, USA | Cat#: 5067–4626 | |
| Commercial assay or kit | 10× Chromium Controller | 10X Genomics | Cat #: 120263 | |
| Commercial assay or kit | Chromium Single Cell v3 reagent kit | 10X Genomics | Cat #: PN-100009 | |
| Software, algorithm | FlowJo | TreeStar | FlowJo 10.6 RRID:SCR_008520 | |
| Software, algorithm | GraphPad Prism | GraphPad Software | GraphPad 9.0 RRID:SCR_002798 | |
| Strain, strain background (mouse) | C57BL/6J | The Jackson Laboratory | Stock Nr. 000664 RRID:IMSR_JAX:000664 | |
| Strain, strain background (mouse) | B6.SJL-*Ptprc^a Pepc^b*/BoyJ | The Jackson Laboratory | Stock Nr. 002014 RRID:IMSR_JAX:002014 | |
| Strain, strain background (mouse) | $Kit^{MerCreMer}$/Rosa26-LSL-eYFP (called $Kit^{MerCreMer}$/R26) | Nanyang Technological University, Singapore *Sheng et al., 2015* | | |
| Strain, strain background (mouse) | Clec9A-DTR | Nanyang Technological University, Singapore *Piva et al., 2012* | | |
| Strain, strain background (mouse) | CD207-DTR | SIgN, A*Star, Singapore *Kissenpfennig et al., 2005* | | |
| Strain, strain background (mouse) | DC-SIGN-DTR | Nanyang Technological University, Singapore | Sheng et al., this paper | |
| Strain, strain background (mouse) | DC-SIGN-DTR-Kit$^{MerCreMer}$/R26 | Nanyang Technological University, Singapore | Sheng et al., this paper | |
| Strain, strain background (mouse) | B6.129S2-$Cd207^{tm2Mal}$/J (Lang-EGFP) | The Jackson Laboratory | Stock Nr. 016939 RRID:IMSR_JAX:016939 | |
| Sequenced-based reagent | Ifng_F | This paper | PCR primers | GACAATCAGGCCATCAGCAAC |
| Sequenced-based reagent | Ifng_R | This paper | PCR primers | ACTCCTTTTCCGCTTCCTGAG |
| Sequenced-based reagent | Il6_F | This paper | PCR primers | TGATGGATGCTACCAAACTGG |
| Sequenced-based reagent | Il6_R | This paper | PCR primers | CCAGGTAGCTATGGTACTCCAGA |
| Sequenced-based reagent | Tnfa_F | This paper | PCR primers | AATTCGAGTGACAAGCCTGTAG |
| Sequenced-based reagent | Tnfa_R | This paper | PCR primers | TTGAGATCCATGCCGTTGG |
| Sequenced-based reagent | Il1b_F | This paper | PCR primers | GGGCCTCAAAGGAAAGAATC |
| Sequenced-based reagent | Il1b_R | This paper | PCR primers | TTCTTCTTTGGGTATTGCTTGG |
| Sequenced-based reagent | Vegfa_F | This paper | PCR primers | GCAGCTTGAGTTAAACGAACG |

*Continued on next page*

*Continued*

| Reagent type (species) or resource | Designation | Source or reference | Identifiers | Additional information |
|---|---|---|---|---|
| Sequenced-based reagent | Vegfa_R | This paper | PCR primers | GGTTCCCGAAACCCTGAG |
| Sequenced-based reagent | HBEGF_F | This paper | PCR primers | ATGACCACACAACCATCCTG |
| Sequenced-based reagent | HBEGF_R | This paper | PCR primers | CCAGCAGACAGACAGATGACA |
| Sequenced-based reagent | cd209a_F | This paper | PCR primers | CCAAGAACTGACCCAGTTGAA |
| Sequenced-based reagent | cd209a_R | This paper | PCR primers | CTTCTGGGCCACAGAGAAGA |
| Sequenced-based reagent | Actb_F | This paper | PCR primers | AAGGCCAACCGTGAAAAGAT |
| Sequenced-based reagent | Actb_R | This paper | PCR primers | CCTGTGGTACGACCAGAGGCATACA |

## Mice

C57BL/6J and B6.SJL-*Ptprc*[a] *Pepc*[b]/BoyJ (B6 CD45.1) were obtained from The Jackson Laboratory (USA). *Kit*[MerCreMer]/Rosa26-LSL-eYFP (called *Kit*[MerCreMer]/R26) and Clec9A-DTR mice were generated as previously described (*Piva et al., 2012*; *Sheng et al., 2015*). Kit[MerCreMer]/R26 mice were backcrossed with DC-SIGN-DTR mice to obtain DC-SIGN-DTR-Kit[MerCreMer]/R26 mice. B6.129S2-*Cd207*[tm2Mal]/J mice were bred and housed at the Malaghan Institute of Medical Research (Wellington, New Zealand). CD207-DTR mice were obtained from the Singapore Immunology Network (SIgN; A*Star, Singapore).

DC-SIGN DTR mice were generated as follows: the IRES-DTR fusion gene was inserted into the 3'-UTR region of the DC-SIGN gene locus on BAC RP24-306K4; the gene targeting vector was then retrieved from the modified BAC (*Figure 7—figure supplement 1A*). The gene targeting vector was linearized and electroporated into Balb/C embryonic stem (ES) cells and correctly recombined ES colonies were selected by PCR. Gene targeted ES cells were injected into C57BL/6 blastocysts and transferred into the oviduct of a pseudo-pregnant mother. F0 male chimera mice were mated with F1 Balb/C females to obtain F1 Balb/C DC-SIGN DTR mice; these mice were then backcrossed to C57BL/6 for 12 generations to generate C57BL/6 DC-SIGN DTR mouse used in this study.

All mice, with the exception of B6.129S2-*Cd207*[tm2Mal]/J mice, were bred and maintained in the specific pathogen-free animal facility of the Nanyang Technological University (Singapore). All studies involving mice in Singapore were carried out in strict accordance with the recommendations of the National Advisory Committee for Laboratory Animal Research and all protocols were approved by the Institutional Animal Care and Use Committee of the Nanyang Technological University (ARF-SBS/NIE A-0133; A-0257; A0126, A-18081). For animal work performed in New Zealand, experimental protocols were approved by the Victoria University of Wellington Animal Ethics Committee and performed in accordance with institutional guidelines.

## TAM-inducible fate mapping mouse models

Kit[MerCreMer]/R26 and DC-SIGN-DTR- Kit[MerCreMer]/R26 fate mapping mice were used to monitor the turnover rates of distinct skin-related DC subpopulation subsets. Upon TAM injection, the YFP label will be induced in all c-kit-expressing cells, predominantly residing in the BM, and these cells will retain the YFP label once they left the BM and seeded into the periphery. Each mouse was administered 4 mg TAM (Sigma-Aldrich, St. Louis, MO, USA) for 5 consecutive days by oral gavage for adult labelling, as previously described (*Sheng et al., 2015*). Pregnant mice (E7.5) were injected once with 16 mg TAM for embryo labelling.

## DT injection

DC-SIGN-DTR[pos] and DC-SIGN-DTR[neg] mice were injected intraperitoneally (i.p.) with 20 ng/g DT (Sigma-Aldrich) to deplete DC-SIGN-expressing cells. Two different DT injection protocols were

used (Figure 7A). For the short-term depletion protocol, mice were injected i.p. at day −2 and −1 before collection of tissues. For the long-term protocol, DT was injected once a week over 5 weeks prior tissue collection.

### Generation of BM chimeras

Chimeric mice were generated by irradiating recipient C57BL/6 or DC-SIGN-DTR mice (CD45.2$^+$) with two doses of 550 cGy, 4 hr apart. Then, 10$^6$ B6.Ly5.1 (CD45.1$^+$) BM cells were injected intravenously (i.v.), 24 hr after treatment. The mice were allowed to recover from 1 to 4 months before analysis.

### Isolation of epidermal, dermal, and LN cells

Mouse ears were cut and separated into dorsal and ventral halves using fine forceps. Both the dorsal and ventral halves (with the epidermis side facing upwards) were incubated for 1 hr at 37°C in 1 ml IMDM (Thermo Fisher Scientific, Waltham, MA, USA) medium containing 1 U/ml Dispase II (Thermo Fisher Scientific). The epidermis and dermis were separated using fine forceps, cut into small pieces and digested for another 1 hr at 37°C in 1 mg/ml Collagenase D (Roche, Basel, Switzerland). To obtain single-cell suspensions, the digested tissue was passed through a 40 mm cell strainer. To process skin-draining auricular LNs, the dissected LNs were minced and incubated in 1 mg/ml collagenase D for 60 min at 37°C.

### Antibodies

The following antibodies were used: anti-mouse CD45 (30-F11), anti-mouse CD11b (M1/70), anti-mouse F4/80 (BM8), anti-mouse Ly6c (HK1.4), anti-mouse CD11c (N418), anti-mouse I-A/I-E (M5/114.15.2), anti-mouse CD103 (2E7), anti-mouse CD326 (G8.8), anti-mouse CD207 (4C7), anti-mouse CD45.1 (A20), anti-mouse CD45.2 (104). They were purchased all from Biolegend (San Diego, CA, USA). Anti-mouse CD45 microbeads from Milteny (Bergisch Gladbach, Germany). All antibodies were used for extracellular stainings with the exception of the anti-CD207 Ab which was used for intracellular labelling after have fixed and permeabilized the cells with 2% paraformaldehyde and 0.05% saponin, respectively.

### Flow cytometry analysis of skin-related DC subpopulations

Single-cell epidermal, dermal, or LN tissue suspensions were pre-incubated with 10 mg/ml anti-Fc receptor antibody (2.4G2) on ice for 20 min. Then, the suspensions were further incubated with fluorochrome-labelled antibodies at 4°C for 20 min, before being washed and re-suspended in PBS/2% FCS for analysis on a five-laser flow cytometer (LSR Fortessa; BD Bioscience, San Jose, CA, USA). The data were analysed with FlowJo software (TreeStar, Ashland, OR, USA) and UMAP analysis was performed using the FlowJo UMAP plugin.

### scRNA-seq analysis

Immune cells were enriched using anti-mouse CD45 microbeads from dermal single-cell suspension. Briefly, enriched CD45$^+$ dermal cells were loaded into chromium microfluidic chips with v3 chemistry and barcoded with a 10× Chromium Controller (10X Genomics, Pleasanton, CA, USA). RNA from the barcoded cells was subsequently reverse-transcribed and sequencing libraries constructed with reagents from a Chromium Single Cell v3 reagent kit (10X Genomics) according to the manufacturer's instructions. Library sequencing was performed at Novogene Co., Ltd (Tianjin Novogene Technology Co., Tianjin, China) with Illumina HiSeq 2000 according to the manufacturer's instructions (Illumina, San Diego, CA, USA).

### Single-cell data analysis

FastQC was used to perform basic statistics on the quality of the raw reads. Raw reads were demultiplexed and mapped to the reference genome by 10X Genomics Cell Ranger pipeline using default parameters. All downstream single-cell analyses were performed using Cell Ranger and Seurat unless mentioned specifically. In brief, for each gene and each cell barcode (filtered by Cell Ranger), unique molecule identifiers were counted to construct digital expression matrices. Secondary filtration for

Seurat analysis: a gene with expression in more than three cells was considered as expressed and each cell was required to have at least 200 expressed genes.

## RNA-seq analysis

All mouse RNAs were analysed using an Agilent Bioanalyser (Agilent, Santa Clara, CA, USA). The RNA Integrity Number ranged from 3.4 to 9.3, with a median of 8.2. cDNA libraries were prepared from a range of 18, 24.2, 68, and 100 ng total RNA starting material using the Ovation Universal RNA-seq system. The length distribution of the cDNA libraries was monitored using a DNA High Sensitivity Reagent Kit on an Agilent Bioanalyser. All 11 samples were subjected to an indexed paired-end sequencing run of 2 × 100 bp on an Illumina Novaseq 6000 system (Illumina, San Diego, CA, USA).

The paired-end reads were trimmed with trim_galore1 (option: -q 20 –stringency 5 –paired). The trimmed paired-end reads were mapped to the Mouse GRCm38/mm10 reference genome using the STAR2 (version 2.6.0a) alignment tool with multi-sample two-pass mapping. Mapped reads were summarized to the gene level using featureCounts3 in the subread4 software package (version 1.4.6-p5) and with gene annotation from GENCODE release M19. DESeq25 was used to analyse differentially expressed genes, and significant genes were identified with Benjamini-Hochberg adjusted p-values<0.05. DESeq2 analysis was carried out in R version 3.5.2.

For functional analysis, hierarchical clustering based on Euclidean distance and complete linkage, was performed using the R 'pheatmap' package. PCA was performed using the R 'prcomp' package. The first two principal components were analysed on a multidimensional scatterplot that was created using the R 'scatterplot 3D' function.

## Preparation and staining of epidermal sheets

DC-SIGN-DTR[neg] and DC-SIGN-DTR[+] mice were treated for 2 days with DT. Ears were collected and split into dorsal and ventral halves and subsequently incubated with 3.8% ammonium thiocyanate (Sigma-Aldrich) in PBS for 20 min at 37°C. Epidermal and dermal sheets were separated and fixed in ice-cold acetone for 15 min. Then, the epidermal sheets were pre-incubated with 10 mg/ml anti-Fc receptor antibody (2.4G2) on ice for 20 min and subsequently stained with FITC-labelled anti-MHC class II antibody for a further 30 min on ice for LC visualization.

## Quantitative real-time PCR

Ears were harvested and immediately homogenized in TRIzol reagent (Thermo Fisher Scientific). Total RNA was subsequently purified using the RNAsimple Total RNA kit (Tiangen Biotech Ltd, Beijing, China). Real-time PCR was performed according to the manufacturer's instructions using the Primer design Precision FAST protocol (Primerdesign Ltd, Cambridge, UK).

## In vitro maturation of LCs

F4/80[hi]CD326[+] LCs were isolated from pooled murine epidermis sheets and purified by cell sorting (purity >90%). $5 \times 10^4$ LCs were immediately used for RNA processing, the remaining $5 \times 10^4$ LCs were cultured in a 96-well round bottom plate for 16 hr in presence of 20 ng/ml GM-CSF and 2 hr/ml LPS and processed the next day for RNA isolation.

## Induction of CHS

WT and DC-SIGN DTR mice were injected with DT and 2 days later were sensitized with 1% DNFB dissolved in an acetone and olive oil mixture (4:1, v/v). DT injection was repeated for 7 days every 3–4 days to maintain the LC pool ablated. The ears of both WT and DC-SIGN DTR mice were challenged with 0.5% DNFB. Ear swelling was measured daily for 12 consecutive days post challenge. Another mouse group was sacrificed at day 3 post challenge for qPCR analysis and for dermal T-cell response analysis.

## Induction of tolerance to CHS

WT, CD207-DTR, Clec9A-DTR, and DC-SIGN DTR mice were injected with DT every 3–4 days over a period of 20 days to maintain the depletion of the target cells (CD207-DTR: LC, cDC1, and LC[like] cells; Clec9A-DTR: cDC1 and DC-SIGN-DTR: LC and CD11b[hi] cells). All mouse strains were tolerized

with a 100 µl volume of 1% DNTB (Sigma-Aldrich) in an acetone and olive oil mixture (AOO) (4:1, v/v), administered epicutaneously to the shaved abdomen. One group of WT mice was painted only with AOO as a control. After 7 days, all mouse strains were sensitized by skin painting the dorsal side of the ears with 0.5% DNFB (Sigma-Aldrich) (25 µl in AOO). Another group of mice was further ear-challenged 5 days later with 0.1% DNFB (4 µl in AOO), and ear swelling was measured using a digital caliper (Mitutoyo, Kanagawa, Japan) over the course of 6 days. In a second group of mice, the draining LNs were harvested at day 5 post tolerization/sensitization.

### Analysis of T-cell responses

To determine the capacity of $CD8^+$ T-cells to secrete IFN-γ, isolated cells were stimulated in a round-bottom 96-well culture plate (Corning, Corning, NY, USA) with 10 ng/ml phorbol 12,13-dibutyrate (PMA, Sigma-Aldrich) and 1 mg/ml Ionomycin (Sigma-Aldrich) in complete IMDM for 3 hr followed by an additional 2 hr incubation with 10 µg/ml Brefeldin A (Sigma-Aldrich) at 37˚C. The cells were then harvested and stained for CD3 and CD8, fixed with 2% paraformaldehyde and permeabilized in 0.05% saponin (Sigma-Aldrich) before staining with anti-IFN-γ antibodies. To quantify activated $T_{regs}$, isolated cells were co-stained for CD4 and ICOS, fixed, and permeabilized using a Fix/Perm Buffer Set (eBioscience) before staining with an anti-Foxp3 antibody.

### Statistics

The data represent the means ± SEM or SD, as indicated in the figure legends. GraphPad Prism software was used to display the data and for statistical analysis. Statistical tests were selected based on the appropriate assumptions with respect to data distribution and variance characteristics. All statistical tests are fully described in detail in the figure legends. Samples were analysed by two-tailed Student's $t$-test to determine statistical differences between two groups. A two-way ANOVA with Bonferroni post-test was used to determine the differences between more than two groups. A p-value < 0.05 was considered to be statistically significant. The number of animals used per group is indicated in the figure legends as '$n$'.

## Acknowledgements

The authors would like to thank Monika Tetlak for mouse management, Su I-hsin and Lim Jun Feng Thomas, for technical advice and Insight Editing London for proofreading the manuscript prior to submission. This work was supported by Ministry of Education Tier1 grant awarded to CR and by the National Key R and D Program of China (Grant 2019YFA0803000) assigned to JS. Work at the MIMR was supported by an Independent Research Organisation Work at the MIMR was supported by a Health Research Council of New Zealand Independent Research Organisation grant to the Malaghan Institute of Medical Research awarded to the Malaghan Institute of Medical Research from the Health Research Council of New Zealand.

## Additional information

### Funding

| Funder | Grant reference number | Author |
|---|---|---|
| Ministry of Education - Singapore | Tier1 | Christiane Ruedl |
| Health Research Council of New Zealand | Independent Research Organisation grant | Franca Ronchese |
| National Key R&D Program of China | 2019YFA0803000 | Jianpeng Sheng |

The funders had no role in study design, data collection and interpretation, or the decision to submit the work for publication.

## Author contributions
Jianpeng Sheng, Conceptualization, Formal analysis, Supervision, Funding acquisition, Investigation, Methodology, Writing - original draft; Qi Chen, Xiaoting Wu, Yu Wen Dong, Johannes Mayer, Investigation, Methodology; Junlei Zhang, Lin Wang, Xueli Bai, Tingbo Liang, Yang Ho Sung, Formal analysis; Wilson Wen Bin Goh, Franca Ronchese, Supervision; Christiane Ruedl, Conceptualization, Data curation, Formal analysis, Supervision, Funding acquisition, Validation, Visualization, Writing - original draft, Project administration, Writing - review and editing

## Author ORCIDs
Qi Chen http://orcid.org/0000-0002-0658-7629
Xiaoting Wu http://orcid.org/0000-0002-0281-8717
Johannes Mayer https://orcid.org/0000-0001-6225-7803
Christiane Ruedl https://orcid.org/0000-0002-5599-6541

## Ethics
Animal experimentation: All studies involving mice in Singapore were carried out in strict accordance with the recommendations of the National Advisory Committee for Laboratory Animal Research and all protocols were approved by the Institutional Animal Care and Use Committee of the Nanyang Technological University (ARF-SBS/NIE A-0133; A-0257; A0126, A-18081). For animal work performed in New Zealand, experimental protocols were approved by the Victoria University of Wellington Animal Ethics Committee and performed in accordance with institutional guidelines.

## Decision letter and Author response
Decision letter https://doi.org/10.7554/eLife.65412.sa1
Author response https://doi.org/10.7554/eLife.65412.sa2

# Additional files

## Supplementary files
• Transparent reporting form

## Data availability
All RNA-sequencing data have been deposited in the Gene Expression Omnibus public database under accession number GSE139877. Single cell RNAseq have been deposited into NCBI SRA database with BioProject ID: PRJNA625270.

The following datasets were generated:

| Author(s) | Year | Dataset title | Dataset URL | Database and Identifier |
|---|---|---|---|---|
| Ruedl C, Sheng J | 2021 | Langerhans cell RNAseq | https://www.ncbi.nlm.nih.gov/geo/query/acc.cgi?acc=GSE139877 | NCBI Gene Expression Omnibus, GSE139877 |
| Wang L | 2020 | Langerhans cells do not migrate to draining lymph nodes | https://www.ncbi.nlm.nih.gov/bioproject/PRJNA625270 | NCBI BioProject, PRJNA625270 |

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
