## [Decision Letter]

**Acceptance summary:**

The present study uses innovative approaches to further our knowledge of skin immunobiology. The corresponding results explain the expression of langerin on two fractions of dermal DC (CD103+ and CD103-) observed several years ago. The demonstration that epidermal LC do not contribute to LN populations in the steady state is completely unexpected and raises important question on the in vivo function of the LC-like subset unveiled in the present study.

**Decision letter after peer review:**

Thank you for submitting your article "Fate mapping analysis reveals a novel dermal migratory Langerhans-like cell population" for consideration by *eLife*. Your article has been reviewed by 3 peer reviewers, and the evaluation has been overseen by a Reviewing Editor and Carla Rothlin as the Senior Editor. The following individuals involved in review of your submission have agreed to reveal their identity: Nikolaus Romani (Reviewer #1); Matthew Collin (Reviewer #2).

Essential Revisions:

1. (a) Stain the new LC-like population for SIRP-alpha (as this was an important marker in Sandrine Henri's 2010 analyses) and (b) assess DC-SIGN protein expression in immature and mature epidermal Langerhans cells by FACS. There are no clear data in the literature regarding DC-SIGN expression on mouse Langerhans cells. The expression of DC-SIGN on mouse Langerhans cells will thus constitute a divergence with human Langerhans cells. Fixing these two points require simple FACS analyses.

2. Perform more in-depth analysis of the single cell data to better probe the relationship of the LC-like subset with LC.

3. Provide information on the proliferation of the LC-like subset. To give rise to such a substantial LN population, the LC-like subset is expected to have a fast turnover. Analysis of the single cell data or a FACS experiment documenting cell cycle markers can contribute to solve this important issue.

*Reviewer #1 (Recommendations for the authors (required)):*

I have some questions for clarifications. Most of the answers will be in the authors' existing files. I suggest two limited set of additional simple (FACS) experiments (see my points #3 and 4) to strengthen the conclusions.

1. Some questions and remarks to Figure 1 – flow cytometry analyses. Answers can easily be provided.

1a. For how many experiments are the FACS plots in Figure 1 representative?

1b. Where are – in Figure 1B dermis – the LC-like cells when backgating onto the F4/80 vs.CD11b plot? Just beneath the LC gate? In other words, how high is the level of F4/80 on the LC-like cells?

1c. Figure 1C. Lymph node. F4/80 vs. CD11b. Is this necessary at all? The gate does not really exclude many cells from further analysis. What is the rationale?

2. Figure 2: Single-cell analyses are interesting and the separate population is convincing. In Figure 2C, however, one cannot really see differences between LCs and LC-like cells. Would the difference in F4/80 expression be mirrored in these plots? Could they be shown? Likewise, maturation marker expression would be informative. Looks like these maturation markers are less expressed in the LC-like cells in panel D – correct? Please provide more information on this point.

3. The authors refer to Henri et al., 2010 in the JEM. Indeed, their populations conform to each other nicely. One marker, though, may be of additional importance, epecially since the authors are referring to that work several times. SIRP-α. Henri in Table I has it on "LC in transit" in the dermis. It discriminates these transiting LCs from the other langerin+ dermal populations. It would therefore be highly valuable to know about the status of this surface antigen on the populations in the dermis and the lymph node.

4. DC-SIGN-DTR mice – Figures 6 and S2. This new mouse model is as yet unpublished. Little if nothing is known about DC-SIGN/CD209 expression on mouse Langerhans cells. Here, the authors show "only" mRNA expression. Could it be possible that LC might down-regulate DC-SIGN surface protein expression upon maturation and in vivo migration? If so, this could explain why LC-like cells are spared from the toxic effect of DT. However, if this is not the case, it would reinforce the authors' conclusions. To determine this would be experimentally very easy: DC-SIGN antibody staining of freshly isolated epidermal LC versus LC cultured for 2 or 3 days – no need to enrich, bulk culture together with keratinocytes suffices – counterstain with langerin or MHC-II. I think this is also important to establish since human LC are clearly DC-SIGN-negative.

*Reviewer #2 (Recommendations for the authors (required)):*

The study uses a range of incisive and complementary technologies to delineate LC-like cells as distinct from epidermal LC, migratory LC and langerin+ cDC1. It is very well written, easy to follow and presents an interesting discussion showing how the current work fits with previous descriptions of murine dermal DC.

1. The dual origin of the LC-like cell (page 7 and figure 4) was inferred from BM chimera experiments in which there was incomplete turnover to donor origin within 4 months. However, this is completely consistent with simple slow kinetics as then demonstrated in the next section (page 8 and figure 5). I do not think that 50% turnover at 4 months following transplantation supports a 'dual origin' if one is considering HSC origin. The language 'dual origin' is confusing because this terminology is in use to describe primitive yolk sac and definitive HSC contributions to LC and macrophage populations. The authors do explain this in the discussion that their use of dual means 'fetal and adult' but this is again a kinetic rate of replacement issue from HSC rather than a critical distinction between primitive yolk sac and definitive HSC. I make this point also because the preceding section (page 6 and Figure 3) shows that the % yolk sac contribution to LC-like cells matches that of HSC-derived DC and is distinct from LC with their small yolk sac contribution. In this sense, the LC-like cell has the same single origin from HSC as other DC. It is just replaced at a slower rate.

2. Proliferation is a major variable that is missing from the data. Could the authors derive this from the single cell RNAseq? In situ proliferation possibly accounts for slow replacement kinetics of LC-like cells yet their substantial contribution to the LN populations. If they are not continually supplied by in situ proliferation in the tissues then presumably LC-like cells must also have an extended life in the LN, compared with other DC?

3. The authors could also do more with their single cell data to define the differences between LC-like and LC. A host of upregulated genes is depicted in Figure 2 in addition to DC-SIGN and TGFB2 but do these define certain pathways or transcription factor signatures?

4. Relate to this is it possible to probe similarities or differences between LC-like cells and other DC? Specifically, the data in 2A could be re-clustered taking only LC and DC which might potentially resolve cDC1 and cDC2 within DC/mono. This is relevant because the discussion about the equivalent human DC misses several studies on langerin expression in cDC2 detectable as mRNA in the blood, in cDC2 inhabiting the dermis and inducible to the same level as LC with TSLP/GM-CSF and TGFb. E.g PMID 25114264, 25352125, 25516751. LC-like cells may be directly comparable with human langerin+ DC (a cDC2) that are neither LC nor cDC1 (25516751). If there is insufficient resolution in the SC RNAseq data then another 6 mice/one lane of 10X might suffice. Without wanting to speculate too much, this relationship between cDC2 and LC may indicate a latent precursor function of cDC2 in humans and in mice (the non-monocyte LC precursor observed in several murine studies).

5. Given that there are about 3-4 times as many migratory LC as LC-like cells in the dermis (Figure 2A) and LC do not appear in the LN then where do they go?

---

## [Author Response]

Essential Revisions:1. (a) Stain the new LC-like population for SIRP-alpha (as this was an important marker in Sandrine Henri's 2010 analyses) and (b) assess DC-SIGN protein expression in immature and mature epidermal Langerhans cells by FACS. There are no clear data in the literature regarding DC-SIGN expression on mouse Langerhans cells. The expression of DC-SIGN on mouse Langerhans cells will thus constitute a divergence with human Langerhans cells. Fixing these two points require simple FACS analyses.

In a new Figure 1- figure 1A supplement, we have included flow cytomentry stainings showing Sirpa (1) and DC-SIGN (2) expression profiles in epidermal LC and dermal LC as well as in dermal LC^like^ and the other three “classical” DC subpopulations.

1. Differently than Henri’s results, dermal LC^like^ cells express Sirpa although at lower levels than the dermal LC counterparts.

2. Although flow cytometry analysis does not visualise DC-SIGN on the cell surface of LCs, our scRNAseq and qPCR analysis show the presence of DC-SIGN specific mRNA which drives the depletion of LCs in the DC-SIGN-DTR mouse. See also our answer to point 4 (Reviewer #1). We have included and discussed these new data in our revised manuscript

2. Perform more in-depth analysis of the single cell data to better probe the relationship of the LC-like subset with LC.

To elucidate the relationship between LC and LC^like^ cells we have included an additional scRNAseq analysis for DC/LC/macrophage specific transcription factors as well as specific myeloid receptors expressed by LCs (Figure 3 and Figure 3- figure 1 supplement).

3. Provide information on the proliferation of the LC-like subset. To give rise to such a substantial LN population, the LC-like subset is expected to have a fast turnover. Analysis of the single cell data or a FACS experiment documenting cell cycle markers can contribute to solve this important issue.

We have added this new information as Figure 6- figure 1 supplement. Please also see our reply to reviewer #2 major point #2.

Reviewer #1 (Recommendations for the authors (required)):I have some questions for clarifications. Most of the answers will be in the authors' existing files. I suggest two limited set of additional simple (FACS) experiments (see my points #3 and 4) to strengthen the conclusions.1. Some questions and remarks to Figure 1 – flow cytometry analyses. Answers can easily be provided.1a. For how many experiments are the FACS plots in Figure 1 representative?

This particular staining was repeated many times over the last years and reproduced also in other laboratories such as at the Malaghan Institute in Wellington, NZ. The LC^like^ cell fraction can be visualized by different gating strategies, one of them is represented in Figure 1. Author response image 1 shows the gating strategy used at the Malaghan Institute giving the same result.

**Author response image 1. sa2fig1:** 

1b. Where are – in Figure 1B dermis – the LC-like cells when backgating onto the F4/80 vs.CD11b plot? Just beneath the LC gate? In other words, how high is the level of F4/80 on the LC-like cells?

To clarify this point (down-regulation of F4/80 and CD11b on LC^like^ DCs), we have not included a new Figure 1—figure supplement 2. Here we have used a Langerin eGFP mouse and backgated the GFP positive cells obtained from epidermis, dermis and LNs. Clearly the dermal CD326^+^CD207^+^ cells consists of two subpopulations (one F4/80^high^CD11b^high^ and a F4/80^int^CD11b^int^)

1c. Figure 1C. Lymph node. F4/80 vs. CD11b. Is this necessary at all? The gate does not really exclude many cells from further analysis. What is the rationale?

This F4/80 versus CD11b dot plot was included to make sure to gate out any possible F4/80^hi^ cell, although we agree they are almost not present in the LN. As advised, we have taken away this first level of back-gating.

2. Figure 2: Single-cell analyses are interesting and the separate population is convincing. In Figure 2C, however, one cannot really see differences between LCs and LC-like cells. Would the difference in F4/80 expression be mirrored in these plots? Could they be shown? Likewise, maturation marker expression would be informative. Looks like these maturation markers are less expressed in the LC-like cells in panel D – correct? Please provide more information on this point.

We have included a violin plot for the F4/80 (*Adgre1*) expression (new Figure 3, figure 1 supplement). Clearly lower mRNA expression in dermal LC^like^ cells mirrors the lower F4/80 surface expression analysed by flow cytometry. We have added also CD80 and CD86 surface expression on epidermal LCs, dermal LCs and dermal LC^like^ (new Figure 1, figure supplement 1A)

3. The authors refer to Henri et al., 2010 in the JEM. Indeed, their populations conform to each other nicely. One marker, though, may be of additional importance, epecially since the authors are referring to that work several times. SIRP-alpha. Henri in Table I has it on "LC in transit" in the dermis. It discriminates these transiting LCs from the other langerin+ dermal populations. It would therefore be highly valuable to know about the status of this surface antigen on the populations in the dermis and the lymph node.

We have now included the expression profile for SIRP-alpha for epidermal LC and for all five dermal DC subpopulations (new Figure 1, figure supplement 1 A).

4. DC-SIGN-DTR mice – Figures 6 and S2. This new mouse model is as yet unpublished. Little if nothing is known about DC-SIGN/CD209 expression on mouse Langerhans cells. Here, the authors show "only" mRNA expression. Could it be possible that LC might down-regulate DC-SIGN surface protein expression upon maturation and in vivo migration? If so, this could explain why LC-like cells are spared from the toxic effect of DT. However, if this is not the case, it would reinforce the authors' conclusions. To determine this would be experimentally very easy: DC-SIGN antibody staining of freshly isolated epidermal LC versus LC cultured for 2 or 3 days – no need to enrich, bulk culture together with keratinocytes suffices – counterstain with langerin or MHC-II. I think this is also important to establish since human LC are clearly DC-SIGN-negative.

We have never claimed that LCs are expressing DC-SIGN on their cell surface and, in fact, we could never see a positive staining with all tested DC-SIGN/CD209 antibodies (now shown in Figure 1—figure supplement 1 B). However our scRNA seq as well as qPCR data show that *Cd209a* is expressed at mRNA levels in LCs. Although the DC-SIGN is undetectable on the LC cell surface the mRNA expression levels are still enough to drive the expression of hHB-EGF, the engineered ligand for the Diphteria toxin. Relevant qPCR data are now added to the Figure 7—figure supplement 1 C. This result explains why LCs are depleted in this DC-SIGN DTR- transgenic mouse model although they do not express the receptor on their cell surface.

To address the raised point of DC-SIGN downregulation upon maturation, we have sorted F4/80^hi^/EPCAM^+^ LCs from epidermis of 20 pooled ear preparations and cultivated them overnight in a 96-well round bottom plate in presence of 20 ng/ml GM-CSF and 2 ng/ml LPS and extracted the RNA the subsequent day. Same sorting was repeated for the unstimulated LCs. qPCR analysis showed that “activated” LCs did not down regulate DC-SIGN (*Cd209a*) mRNA and therefore we can exclude a down-regulation of this receptor upon maturation.

Reviewer #2 (Recommendations for the authors (required)):The study uses a range of incisive and complementary technologies to delineate LC-like cells as distinct from epidermal LC, migratory LC and langerin+ cDC1. It is very well written, easy to follow and presents an interesting discussion showing how the current work fits with previous descriptions of murine dermal DC.1. The dual origin of the LC-like cell (page 7 and figure 4) was inferred from BM chimera experiments in which there was incomplete turnover to donor origin within 4 months. However, this is completely consistent with simple slow kinetics as then demonstrated in the next section (page 8 and figure 5). I do not think that 50% turnover at 4 months following transplantation supports a 'dual origin' if one is considering HSC origin. The language 'dual origin' is confusing because this terminology is in use to describe primitive yolk sac and definitive HSC contributions to LC and macrophage populations. The authors do explain this in the discussion that their use of dual means 'fetal and adult' but this is again a kinetic rate of replacement issue from HSC rather than a critical distinction between primitive yolk sac and definitive HSC. I make this point also because the preceding section (page 6 and Figure 3) shows that the % yolk sac contribution to LC-like cells matches that of HSC-derived DC and is distinct from LC with their small yolk sac contribution. In this sense, the LC-like cell has the same single origin from HSC as other DC. It is just replaced at a slower rate.

We amended accordingly and have taken away dual origin.

2. Proliferation is a major variable that is missing from the data. Could the authors derive this from the single cell RNAseq? In situ proliferation possibly accounts for slow replacement kinetics of LC-like cells yet their substantial contribution to the LN populations. If they are not continually supplied by in situ proliferation in the tissues then presumably LC-like cells must also have an extended life in the LN, compared with other DC?

Skin (epidermis and dermis) host distinct DC subpopulations in specific niches where innate and environmental factors control DC life span, turn-over, in situ proliferation and their numbers.

As suggested by the reviewer, cell cycle analysis was performed based on the scRNA-seq data, LC^like^ cell did show a higher portion of proliferative cells compared to other macrophage and DC subsets within dermis (new Figure 6, figure supplement 1). LCs are known as long lived tissue resident macrophages of epidermis showing only few actively dividing cells. DCs, such as cDC1, TN DC and CD11b^hi^ DCs are short lived cells with fast turn-over (Ruedl et al., 2000), therefore compensation of DCs in the tissue depends on the contribution of their BM progenitors. While LC^like^ cells with a slower turnover kinetics exhibit higher proliferating capability to refill the emigration gap.

3. The authors could also do more with their single cell data to define the differences between LC-like and LC. A host of upregulated genes is depicted in Figure 2 in addition to DC-SIGN and TGFB2 but do these define certain pathways or transcription factor signatures?

We have included now a detailed profiling of different transcription factors and representative myeloid and DC/LC-related molecules comparing 8 distinct dermal LC, LC^like^ and other DC and macrophage subpopulations (Figure 3, figure supplement 1).

4. Relate to this is it possible to probe similarities or differences between LC-like cells and other DC? Specifically, the data in 2A could be re-clustered taking only LC and DC which might potentially resolve cDC1 and cDC2 within DC/mono. This is relevant because the discussion about the equivalent human DC misses several studies on langerin expression in cDC2 detectable as mRNA in the blood, in cDC2 inhabiting the dermis and inducible to the same level as LC with TSLP/GM-CSF and TGFb. E.g PMID 25114264, 25352125, 25516751. LC-like cells may be directly comparable with human langerin+ DC (a cDC2) that are neither LC nor cDC1 (25516751). If there is insufficient resolution in the SC RNAseq data then another 6 mice/one lane of 10X might suffice. Without wanting to speculate too much, this relationship between cDC2 and LC may indicate a latent precursor function of cDC2 in humans and in mice (the non-monocyte LC precursor observed in several murine studies).

We have analysed our scRNAseq data of dermal LC/LC^like^/DC and macrophage subpopulations for transcription factors involved in DC and macrophage development and included the results in new Figure 3 and Figure 3—figure supplement 1. Clearly LC and LC^like^ cells a very similar although some of the TFs (*Id2, Klf4, Runx2 and 3, Stat3*), are higher expressed in LC^like^ cells, whereas *pu.1* is higher expressed in LC. TFs highly expressed in conventional DCs such as *Flt3* (cDC1, cDC2, TN), *Irf 8* (cDC1), *Zeb2* (cDC2) are expressed in LC^like^ cells but not at the level of DCs, therefore we postulate their independent lineage. Obviously further experiments will be required using specific TF KO mice (currently not available in our laboratory) to profile the TF network involved in the LC^like^ development. We have added this in the discussion part and cited the relevant published work about human langerin+ cDC2 cells.

5. Given that there are about 3-4 times as many migratory LC as LC-like cells in the dermis (Figure 2A) and LC do not appear in the LN then where do they go?

Currently we do not know and can only speculate. We favour the hypothesis that LCs, at least at steady state, similar to other tissue-resident macrophages, are sessile and act locally in the skin and our in vivo functional data show that LC are one of the main immunosuppressive regulators within the skin.